# Application of Low-Cost Sensors for Building Monitoring: A Systematic Literature Review

**Behnam Mobaraki** [1,*]**, Fidel Lozano-Galant** [1]**, Rocio Porras Soriano** [2]  **and Francisco Javier Castilla Pascual** [3]

1 Department of Civil and Building Engineering, Universidad de Castilla-La Mancha (UCLM), 13071 Ciudad Real, Spain; fidel.lozanogalant@uclm.es
2 Department of Applied Mechanics and Projects Engineering, Universidad de Castilla-La Mancha (UCLM), 13071 Ciudad Real, Spain; Rocio.Porras@uclm.es
3 Department of Civil and Building Engineering, Universidad de Castilla-La Mancha (UCLM), 16071 Cuenca, Spain; fcojavier.castilla@uclm.es
* Correspondence: behnam.mobaraki@uclm.es

**Abstract:** In recent years, many scholars have dedicated their research to the development of low-cost sensors for monitoring of various parameters. Despite their high number of applications, the state of the art related to low-cost sensors in building monitoring has not been addressed. To fill this gap, this article presents a systematic review, following well-established methodology, to analyze the state of the art in two aspects of structural and indoor parameters of buildings, in the SCOPUS database. This analysis allows to illustrate the potential uses of low-cost sensors in the building sector and addresses the scholars the preferred communication protocols and the most common microcontrollers for installation of low-cost monitoring systems. In addition, special attention is paid to describe different areas of the two mentioned fields of building monitoring and the most crucial parameters to be monitored in buildings. Finally, the deficiencies in line with limited number of studies carried out in various fields of building monitoring are overviewed and a series of parameters that ought to be studied in the future are proposed.

**Keywords:** low-cost sensor; building monitoring; state of the art; systematic literature review; microcontroller; communication protocol





## 1. Introduction

Building monitoring has become a matter of concern for engineers and architects as structures and building envelopes should meet the safety, habitability, and sustainability requests during the service life [1,2]. Accordingly, engineers decided to control performance of the buildings by monitoring two main parameters: (1) Structural parameters: in terms of structural system identification (SSI), some numerical approaches have been developed for inferring mechanical parameters of structures modeled with 1D elements (such as steel and concrete buildings, cable stayed bridges, trusses, and frames) [3–5], 2D elements (such as tunnels, culverts, and dams) [6], and 3D elements [7]. In terms of deriving parameters of structures modeled with 3D elements, various approaches have been presented in the literature [8,9]. For instance, Mobaraki and Vaghefi developed 3D finite element models to measure peak pressure and the peak particle velocity (PPV), at critical points of 4 cross-sectional shape of tunnel (box shape, circular shape, horseshoe shape, and semi-ellipse shape) [10]. Lyapin et al. introduced a methodology for monitoring buried structures of arbitrary cross-section, located in layered media, affected by different external dynamic loads [11]. They developed and analytical approach for determination of resonance zones and dynamic response of structures. In addition, they conducted an experimental study for measuring acceleration at the ceiling center point of an underground pedestrian road in Russia. A review of research carried out in the field of SSI was presented by Sirca Jr. and Adeli [12]. The methodology for identification of

modal parameters in historical building using stochastic subspace identification algorithm carried out by Bianconi et al. [13]. Markiewicz et al. utilized a geo-localization approach to specify the location of the structural monitoring system that allowed to geo-reference the measurements carried out by the sensors. This analysis is useful for data processing related to the monitored structure and its features [14]. Applicability of strain gauge to real time monitoring of wood behavior carried out was carried out by Anaf et al. in a church where a new heating system was installed [15]. For structural safety assessment, as well as calibration of the structural models, measurements of acceleration, and deflection at different parts of buildings are needed. There are varieties of measurement devices in the market for the measurement of the mentioned parameters. The price of these devices is normally high as they contain expensive sensors, data treatment protective box, power system, and software for data acquisition system. According to the report published by BBC research center, the size of the global sensors market has dramatically increased since 2011 [16]. It can be said that this market was valued at USD 101.9 billion in 2015 and by considering a compound annual growth rate of 11.0% will reach to USD 190.6 billion by 2021. (2) Indoor building parameters: One of the principal indoor parameters to be monitored is the energy efficiency of building [17]. In terms of controlling this parameter, diverse methodologies have been proposed for characterization of thermal parameters of the buildings. Traditionally, it was necessary to core the building envelope, measure thickness of each layer and after assigning conductivity value to each layer, the transmittance parameter of the object could be inferred. To avoid implementation of the destructive approaches to the buildings, several methodologies were developed for thermal parameter diagnosis of the buildings. Functionality of these approaches is normally based on costly measurement of the ambient temperature, surface temperature, and heat flow rate through the object. Accurate estimation of these parameters requires multiple measurement points on the structures. However, due to the expensive instrumentation of these methodologies, only a single measurement point is considered by engineers to determine condition of the structure [18]. Review of the current expensive methodologies for thermal monitoring of the buildings can be found in the article of Teni et al. [19].

In this regard, there are multiple codes for defining a set of guidelines to be considered in different types of building energy performance. Some of the common monitoring standards in Europe are as follows: (1) EN 15232 provides the requirements for the building automation, control, building management functions exerted to various types of building, and effect of different functions on energy performance of buildings. (2) ISO 16739-1:2018 is a standard for building information modeling (BIM). This standard provides BIM exchange format definitions which are needed during the life cycle of buildings and are required by diverse disciplines (e.g., architecture, project management and building service). (3) ISO 6946:2017 for theoretical calculation of the thermal transmittance parameter (U-value) of the buildings, by considering the thicknesses of the wall's compositions (4) ISO 9869: 2014 presents the heat flux meter (HFM) method for calculation of the thermal transmission of the buildings' components under a steady state condition (although this situation cannot be achieved from physical-mathematical point of view). (5) ISO 6781-3:2015 detection of heat, air, and moisture irregularities in buildings by infrared methods. (6) ISO 10456:2007 presents the methods for the calculation of the thermal parameters of the buildings by introducing the thermal conductivity of different materials. Among the explained standards, numbers (1) and (4) present multiple provisions, such as tests duration, frequency of the observations, seasonality of experiments, treatment of the outliers, and verification methods of the observations.

The development of a building monitoring system (BMS) requires the installation of a data acquisition system for real time observation of parameters and a network for storing of the data [20]. Unfortunately, due to the complexity of the sensors, traditional BMS is not an easy task as it entails specialized programing and maintenance of the system. In this way various monitoring systems have been produced by companies for direct supervision of building performance. As depicted in Figure 1, a common monitoring system contains

three main parts. (1) A detection part. This part contains a sensor or data acquisition system to measure the events and changes of a parameter. (2) Data transfer system. This part is used to transmit readings from one place to another through a communication method and (3) storage part. This part consists of a component to keep the digital data.

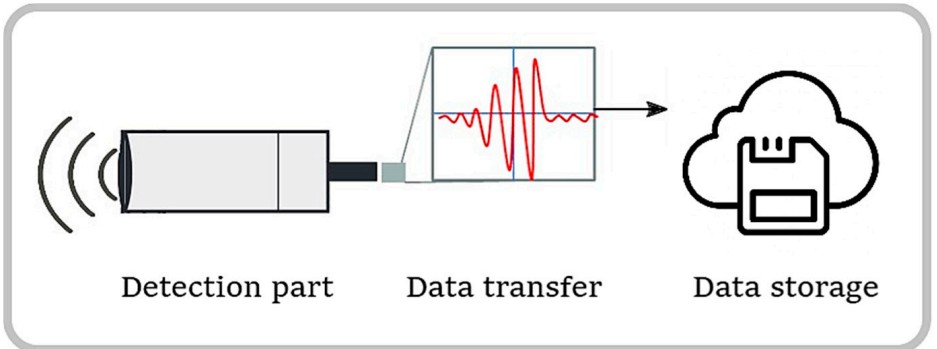

**Figure 1.** Schematic of a data acquisition system.

After saving and processing the data, the principal objective is to assess the workability of the building and safety of the habitants. Recently, diverse technologies have been developed with the purpose of real time monitoring of buildings and safety management of people. In fact, the most flourishing technology for this aim is BIM. This tool is a novel approach to monitor, analyze, design, and manage safety, wherein visualization of the building state is utilized to enable the exchange and interoperability of data. BIM has been adopted by CAD software such as: Autodesk Revit [21], ArchiCAD [22] and Allplan [23]. In 1900s Matilla et al. studied significance of studying the connections between management and safety [24]. Research of Ding et al. indicated that the quantity of the published articles on BIM from the perspective of the safety management is 7%. The other advanced technology that brought great potential for safety management of people is wireless sensor network (WSN). With cheap, low-power consumption, and dynamic networking characteristics, WSNs are good option for collection of environmental and structural information of buildings and return it to visualizing unit for safety management and risk analysis [25]. Cheung et al. integrated BIM and WSN into a single system to visualize the construction site, monitor the safety level through an interface and remove the risk or danger of the gas automatically [26]. They have also carried out safety management processes of underground structures by monitoring the other elements of environmental parameters, such as temperature and humidity. They installed the sensor nodes at different location of the structure, and in any area where an abnormal situation is realized, the BIM model alarms the area and ventilator starts working, automatically.

Table 1 presents a list of common environmental monitoring devices in the market providing the products' name, detection range, accuracy, cost, and the reference of their applications in the literature.

**Table 1.** List of commercial systems in the market for indoor monitoring of buildings.

| Application | Device | Detection Range | Accuracy | Price (€) | Ref. |
|---|---|---|---|---|---|
| | PROTMEX MS6508 | −20 to 60 °C | 1.0 °C | 57 | [27] |
| | REED R6001 | −20 to 60 °C | 0.8 °C | 103 | [28] |
| Temperature | FLUKE 971 | −20 to 60 °C | 0.5 °C | 464 | [29] |
| | EL–USB-2 LASCAR | −35 to 80 °C | 0.5 °C | 57 | [30] |
| | TESTO 435-1 thermocouples class 1 | −50 to 150 °C | 0.2 °C | 1032 | [31] |

**Table 1.** *Cont.*

| Application | Device | Detection Range | Accuracy | Price (€) | Ref. |
|---|---|---|---|---|---|
| | EXTECH EN510 | −100 to 1300 °C | 0.1 °C | 180 | [32] |
| Gas | NDIR | 0–10,000 ppm | 30–200 ppm | 80–550 | [33] |
| | MOSFET | 400–20,000 ppm | 30–100 ppm | 20–2300 | [34] |
| | Electrochemical | 0–1000 ppm | 0–30 ppm | 85–620 | [35] |
| Humidity | Captive sensors | 0–100% RH | 0–5% | 30–180 | [36] |
| | Resistive sensors | 5–90% RH | 1–10% | 30–140 | [37] |
| Airflow | Hot wire Anemometer | 0.1–45 m/s | 1–5% | 45–190 | [38] |
| | Vane Anemometer | 0.25–50 m/s | 1–5% | 30–280 | [39] |

In terms of structural monitoring, there are various tools to measure different parameters such as acceleration, strain, and stresses. Accelerometer is an electromechanical device that determines the acceleration forces imposed to or acting on an object. There are varieties of accelerometer that work based on different mechanism such as: (1) Piezoelectric, this type of accelerometer uses piezoelectric effect of specific material and transform a type of energy into another one and produce electrical signal in response to parameter is being measured. (2) Piezoresistive, this type of accelerometer generates resistance changes in displacement sensor which is part of accelerometer system. This type of accelerometer is the best option for measuring impulse where the amplitude and frequency range are high. (3) Capacitive micro electro mechanical system (MEMS), for construction of this type of accelerometer, MEMS technology is being used and works according to capacitance changes in a seismic mas under acceleration. Table 2 provides examples of the mentioned types of accelerometers in the market (including both and high frequency) providing the mechanism, the name, acceleration range, frequency range, and the references in the literature. The accelerometers utilized for measuring specific human activities, transportation, and mechanical devices must be developed specially for low frequency and high sensitivity, range from 1 to 10 Hz [40]. Tian et al. developed, a piezoelectric accelerometer on n-type single crystal silicon and examined the sensor in terms of maximum stress, natural frequency, and output voltage under an acceleration through the finite element method. The sensitivity of the developed accelerometer was 9 mV/g, the linearity was 0.0205, and the hysteresis was 0.0033 [41]. Another example of low frequency piezoelectric accelerometers in the literature are [42,43]. Liu et al. proposed a novel low frequency fiber bragg grating (FBG) accelerometer with a bended spring plate. The experiments showed the sensitivity of the accelerometer was more than 1000 pm/g when the frequency is within the 0.7 to 20 Hz [44]. Another example of low frequency FBG accelerometers can be found in [45–47]. Zhu et al. designed high resolution, low frequency and low-noise tri-axial digital MEMS accelerometer for monitoring large-scale civil infrastructures [48]. Examples of the other low frequency MEMS accelerometers were presented in [49,50].

**Table 2.** Example of common accelerometers in the market.

| Mechanism | Device | Acceleration Range(g) | Frequency Range (KHz) | Price (€) | Ref. |
|---|---|---|---|---|---|
| | IAC-HiRes-I-03 | ±25 | 0–10 | 2230 | |
| | MS9002 | ±2 | 0–2 | 286 | [51] |
| Capacitive | MS9010 | ±10 | 0–10 | 286 | [52] [53] |
| | MS9050 | ±50 | 0–50 | 286 | [54] |
| | MS9100 | ±100 | 0–100 | 286 | |

**Table 2.** *Cont.*

| Mechanism | Device | Acceleration Range(g) | Frequency Range (KHz) | Price (€) | Ref. |
|---|---|---|---|---|---|
| | MS9200 | ±200 | 0–200 | 571 | |
| Piezoelectric | 3713B112G | ±2 | 0–25 | 2070 | [55] [56] |
| | Dytran 3143D1 | ±50 | 0.0005–3 | 1255 | |
| | Dytran 3093B | ±50 | 0.006–5 | 1255 | |
| | Dytran 3263A14 | ±250 | 0.0005–4 | 1255 | |
| | Dytran 3093M27 | ±500 | 0.0033–3 | 1525 | |
| | Dytran 3093M18 | ±500 | 0.007–5 | 1525 | |
| Piezoresistive | 3501A2020KG | ±20,000 | 0–10 | 960 | [57] |
| | 3503C2060KG | ±60,000 | 0–10 Hz | 6750 | |
| | 3991B1120KG | ±20,000 | 0–10 Hz | 2500 | |

Demands for efficient and low-cost monitoring of the buildings is continuing to increase [58]. This matter can be the first step to reduce uncertainty of building monitoring by increasing the density of measurement points [59]. With the quick increase in micro sensor technology and jutting microcontrollers (e.g., Arduino and raspberry), the quick adoption of low-cost sensor (LCS) for various aspects of building monitoring can be seen in the literature [60]. The example of these studies are monitoring of: temperature [61], humidity [62], light [63], $CO_2$ [64], particle matter [65], airflow [66], occupancy [67], energy consumption [68], electricity and window opening times [69], solar irradiance [70], solar heat flux [71], strain [72], acceleration [73], and for detecting the earthquake [74], triaxial acceleration [75].

There are various review papers in the literature concerning: the approaches for smart monitoring of buildings [76], environmental monitoring sensors in buildings [77], heritage building information modeling [78], BIM-based end-of-lifecycle decision making and digital deconstruction [79], in-situ measurements of the building thermal parameters [80], indoor air quality monitoring system based on Internet of Things (IoT) [81], indoor particle matter monitoring [40], available techniques for monitoring of energy in buildings [19], monitoring the power usage of appliance in buildings [21], monitoring thermal comfort of the habitants based on the IoT paradigm [24], occupancy monitoring for energy saving in commercial buildings [38], and key sensors for monitoring of concrete structures [82]. Bakker et al. focused on the approaches for monitoring of buildings occupancy based on lighting control. They identified how much the occupancy-based lightning systems have been tested, developed, and controlled [83]. Salim et al. conducted a survey on the analysis of the articles in literature focused on modeling of buildings occupant's behavior and the rate of energy consumption at different scales [84]. A review of the optical fiber sensors used for monitoring of multiple parameters in buildings (such as strain, temperature and pressure) can be found in the article of Annamdas [85]. Sun et al. discussed the application of optic sensors, piezoelectric sensors, self-diagnosis fiber reinforced composites, and magnetostrictive sensors for building monitoring [86]. Lynch and Loh reviewed the advantages of using wireless sensors and sensor networks for building monitoring [87].

At the same time, the other authors provided an overview of LCSs for monitoring of a specific parameter in the buildings. Bilro et al. reviewed the potential application of optical sensors based on plastic fibers used for low-cost building monitoring [88]. Barrias et al. presented the theoretical background of distributed optical fiber sensors, as well as multiple application of that in building monitoring [89]. Saini et al. introduced indoor air quality monitoring systems based on the publications reviewed from 4 databases, Web of science, IEEE explore, Science direct, and Pubmed. They also highlighted the most preferred microcontrollers (Arduino UNO, Raspberry Pi, and ESP8266) communication technologies

(Wi-Fi, Bluetooth, and ZigBee), and data storage methods (cloud server and local server) for development of low-cost air quality monitoring systems in the buildings [81]. Karagulian et al. presented a review article dealing with conducted research regarding the performance of LCS for air quality monitoring and at the end they proposed the most cost effective LCS that can be used for this aim [90]. Penza et al. presented a literature review, presenting application of low-cost sensors for monitoring air quality in buildings, offices, schools, street, port, and airports [91]. Spinelle et al. presented a literature review for monitoring of benzene in ambient air and volatile organic compounds using LCSs. They also presented pros and cons of the sensor technologies using for benzene detection [92]. Yang et al. reviewed new technologies for determining the thermal comfort in buildings, based on monitoring of people thermal physiology. They also introduced the pros and cons of both the low-cost and conventional methodologies [93].

Although some literature reviews for low-cost monitoring of a single parameter in buildings have been published, a systematic literature review on the application of LCSs for monitoring of essential parameters (structural as well as the indoor building ones) in the buildings is missing from the current reviews. To fill this gap, this review paper works on reviewing the articles dealing with low-cost monitoring of buildings in the SCOPUS database. In addition, the authors are paying special attention to present various aspects of the main fields of building monitoring (structural and indoor building parameters), the most common microcontrollers for implementation of low-cost monitoring systems, and the preferred data acquisition methods.

This paper is organized as follows. In Section 2, the methods applied for inclusion, exclusion, and classification of the articles are presented and specific objectives of the research are defined. Moreover, the methodology for data processing of the low-cost monitoring devices is introduced. In this section, some research questions are presented to make the systematic review and follow the determined protocol. In Section 3, detailed explanation about various fields of building monitoring and reviewed article are given. In Section 4, presentation of the different types of microcontrollers and communication protocols in the literature are reviewed and the preferred ones for installation of low-cost monitoring devices are presented. In Section 5, the significance of the combination of BIM and LCSs are introduced, and the associated references are given. Section 6, the obtained results, importance of different aspects of building monitoring are discussed and some proposal for future investigation are presented, Finally, in Section 7 some conclusions are drawn.

## 2. Materials and Methods of Classifications

In this section, the origin of the data, methods of evaluation, and classification of LCSs used for building monitoring, are presented. The two main fields of building monitoring are associated with indoor and structural parameters of buildings. These two fields of monitoring that may cover and present serviceability of a building during its service life. This article is wholly focused on the papers publications in the SCOPUS database from 2006 to 2020, as the found articles were published in this period. This systematic literature review was carried out as of February 2021. The initial search strategy implemented to this systematic review paper incudes three main steps. In the first step by implementing a specific search algorithm (Figure 1), "AND" a first group of articles was obtained. In the next step, the found articles were additionally filtered to omit duplicate and irrelevant ones. Following this step, 652 articles were considered inappropriate because of missing information associated with building monitoring, or that they did not obviously discuss low-cost monitoring. In this research the authors were only focusing on the articles written in the English language as it is more demanding than the other ones between the scholars. This exclusion was carried out using "AND NOT" command and resulted in 62 articles for monitoring the indoor parameters of buildings and 37 articles for the structural part of buildings. In the third step, a detailed review of the found articles was carried out in terms of defining the principal focuses of the selected articles, assessing the distribution of them

in different countries of the world, and finally evaluating the types of microcontrollers and communication protocols for installation of the low-cost monitoring system. Figure 2 illustrates a summary of the explained steps carried out to select the relevant publications from the SCOPUS database.

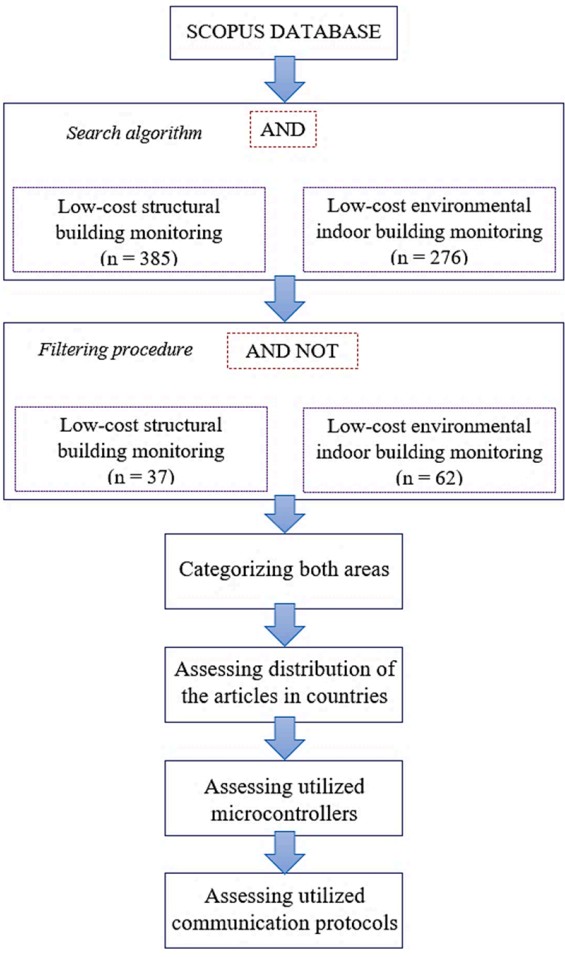

**Figure 2.** Flow diagram for systematic review.

Summary of the studied fields of building monitoring and their defined categories are presented in the Table 3. It can be seen that the indoor parameters of buildings contain controlling the electricity consumption, air quality, thermal comfort and heating, ventilation, and air conditioning (HVAC), and some other special indoor parameters. Structural performance of buildings is also defined by controlling vibration and strain parameters.

**Table 3.** Defined categories for the two main fields of building monitoring.

| Fields of Building Monitoring | Indoor | | | | Structural | |
|---|---|---|---|---|---|---|
| Category | 1 | 2 | 3 | 4 | 5 | 6 |
| Studied groups | Electricity consumption | Air quality | Thermal comfort and HVAC | Others | Vibration | Strain |

### 2.1. Research Question

There are numerous types of LCSs that have been used in the literature for different monitoring projects. However, the principal aim of this article is to present comprehensive information about low-cost building monitoring from different points of view, highlighting

appropriate microcontrollers, and communication protocols. Thus, this article can be served as a comprehensive reference for scholars in promoting low-cost technologies of monitoring all parameters of buildings. To do so, in this subsection, some initial research questions are introduced, and in following sections precise analysis and explanation are given to readers.

1. What are the preferred types of microcontrollers for low-cost monitoring?
2. What are the chosen communication protocols?
3. What are the most important parameters to be monitored in the field of building monitoring?

In fact, question 1 provides information about available types of microcontrollers in the market for installation of the low-cost monitoring system. By presenting the most preferred microcontroller in the literature, it also gives the insight to the readers to choose the most appropriate one for his or her project. Question 2 gives insight into diverse communication protocols available for data acquisition systems. By reviewing the most common one for building monitoring, as well as presenting their pros and cons, readers can have better understanding of the existing options for their installation. Questions 3 gives a proper understanding of crucial parameters in monitoring projects that have priority to all possible aspects of building monitoring.

### 2.2. Data Processing of Low-Cost Monitoring Systems

New achievements in technology of metering devices, wireless communications, and data processing methods, in conjunction with ever increasing number of aged buildings, have led to the development of more efficient monitoring systems. In fact, the principal feature of the current low-cost monitoring devices is that they contain open-source programs. These metering systems provide permission to the users to use the library source of the utilized low-cost sensors, content of the data acquisition system, and criteria for the processing of the measurements. This implies that the users have chance of copying, improving, and developing an algorithm for post-processing of their own measurement data, freely [81]. In the context of building energy retrofitting and monitoring, various scholars developed specific data acquisition systems [87,94,95]. The challenges of simulations, experimental measurements, and post-processing of data in terms of building thermal monitoring was studied by Evangelisti et al. [96]. In the first step, the authors investigated how the disturbing factors might influence the final results of U-value and then they developed a novel post-processing method data analysis called "linear trendline" approach [96]. Taking into account the finite element simulation and in-situ measurements, they received satisfactory results by reducing the difference between the measured and calculated transmittance value. Marquez et al. developed a novel low-cost U-value meter which contains open source software and incorporates a user interface [97]. It also delivers real time monitoring of the data, processes of the information, detecting communication errors and warns of outliers. An example of a building energy management system using self-deployed open source platforms for data processing of multiple variables can be found in [98]. Some authors, in this line, have introduced an open source signal processing algorithm in terms of monitoring structural parameters of buildings [52,59,87,99].

However, majority of the conventional devices in the market in any fields of building monitoring, have already programed by the associated companies and they are based on the close source platforms. This means that only the original authors are able to inspect and improve the developed software. These companies provide the users by the software for post-processing of information and the users will receive the update of the source code, annually. The users must agree that they cannot alter or copy anything from the source code. Examples of these metering devices for measuring thermal transmittance of the building envelope can be find in refs. [18,80].

## 3. Results

The first publications found in the SCOPUS presenting the idea of using LCSs dealing with indoor monitoring of buildings and structural ones are both dated in 1993 (these two articles were not considered in the statistical analysis of this research as they were introducing the idea of utilizing low-cost sensors (e.g., fiber optic sensors) in building monitoring). Measures evaluated the types of low-cost strain optic sensors and he approved the applicability of Bragg grating which had satisfied the requirements for smart structures, due to the developed ratiometric wavelength demolition system [100]. Tsilingris presented a low-cost flat-response, temperature compensated sensor designed for measuring of the radiation transmission, that can be used for underwater uses and solar ponds [101]. Figure 3 illustrates the significant of LCSs in the articles found in SCOPUS database from 2006 to 2020. It can be observed in the figure that since 2006 until 2013 the quantity of the publications associated with the use of LCS for structural monitoring are negligible (27%) and more than 73% of the articles have been published in the last 7 years (from 2014 to 2020). This rebound in terms of projects dealing with indoor monitoring of buildings occurred at the same time as the number of publications has the share of 77.4% in the last 7 years and only 22.6% of them were conducted in the first 8 years.

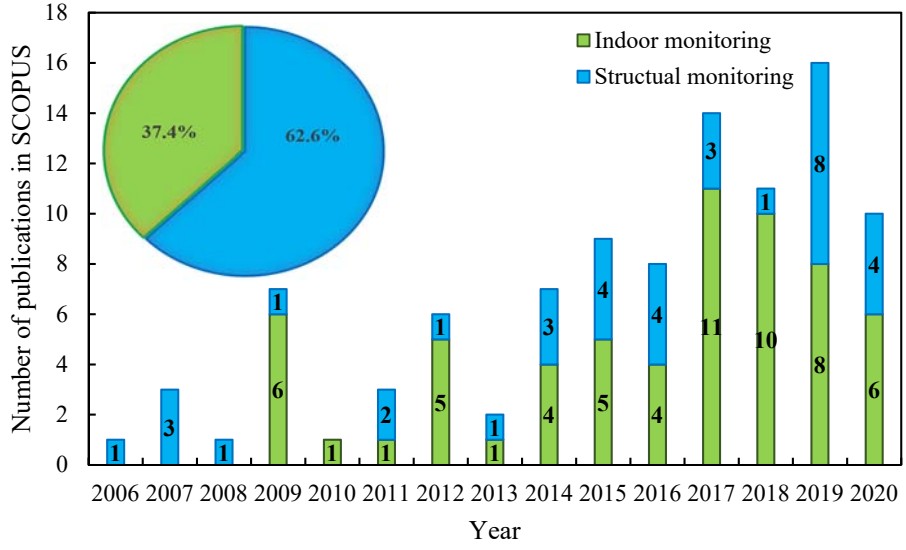

**Figure 3.** Distribution of the reviewed articles from 2006 to 2020 in SCOPUS database.

### 3.1. Indoor Monitoring of Buildings

In this section, the reviewed articles associated with low-cost monitoring of buildings in terms of indoor parameters are presented. This first group of articles can be categorized into four separated categories. Category 1 are a group of the authors that dedicate their studies on the application of LCSs to measure the efficiency of the electrical appliance and lessen the whole energy consumption of the buildings, category 2 are a group of the authors that focus on controlling of the air quality of the buildings through the measurement of the $CO_2$, pollutant emission and aerosol nanoparticles, category 3, are a group of authors that address the thermal comfort and satisfaction of the buildings' habitants by measuring the indoor temperature and humidity in their studies. This category also contains HVAC. Finally, category 4 contains publications dealing with special parameters of building monitoring such as fire and chemical detection, low-cost thermal monitoring of buildings, and the other parameters.

Figure 4 illustrates distribution of the reviewed articles in SCOPUS associated with the above outlined categories. It can be seen among the four categories, the highest number of published articles were dealing with the application of LCSs for thermal comfort monitoring of the building habitants and HVAC (Category-3). This group contains 39% of the publications from 2009 to 2020. The analysis of the Figure 4 proves that concerns with

thermal comfort and indoor satisfaction of the people are significantly growing with time as majority of the publications (75%) in this area are from that last four years. Majority of the electrical energy in the world consumes in the building and industry rather than other sectors such as transport, fishing, and agriculture [102]. Analysis of the Figure 4 shows that distribution of the publications dealing with low-cost control of electricity consumption (category 1) in the building was almost constant during the years (from 2012 to 2020) and no significant change has been occurred. This group of the articles contains 26% of the publications in the field of indoor monitoring of buildings.

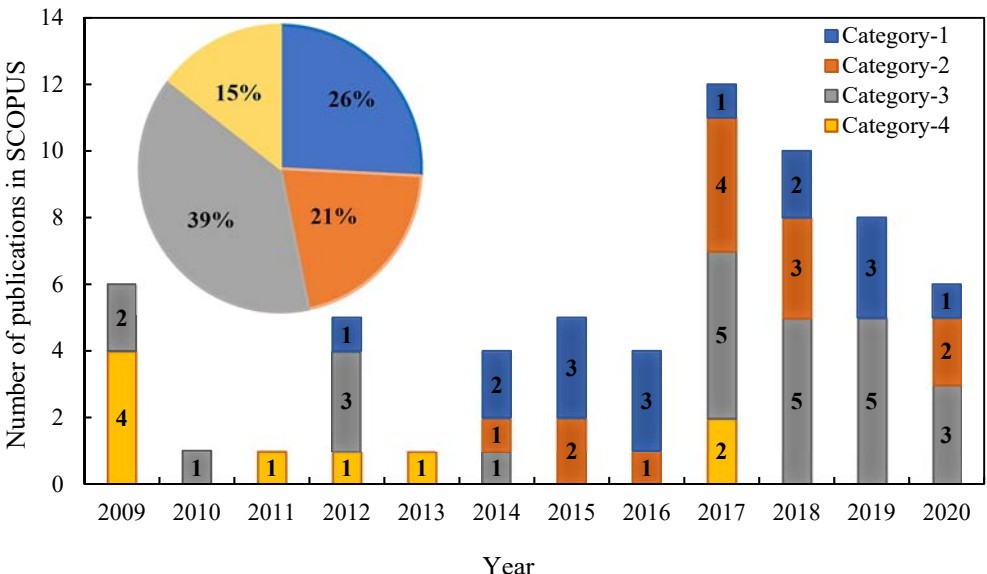

**Figure 4.** Distribution of the reviewed articles associated with the application of LCSs for indoor monitoring of buildings from 2009 to 2020. Category-1: Energy efficiency of the electrical appliance, category-2: controlling of the air quality and the pollutions, category-3: thermal comfort and HVAC, and category-4: other aspects of indoor monitoring of buildings.

In fact, the growth of the industrialization in the world, has caused an increase in pollutant emission in cities. Simultaneously, the demands for the controlling the air quality in buildings is increasing. The category 2 (controlling the indoor air quality and the level of indoor pollution) contains 21% of the whole publications in the SCOPUS database. The last group of the articles has the share of 15% in the whole reviewed article. Those papers described different investigations from the first three groups such as: low-cost controlling of water consumption, low-cost monitoring of the chemical pollution, low-cost fire detection in the buildings, and low-cost thermal parameter diagnosis of building envelopes.

This section of the systematic review (monitoring indoor parameters of buildings) contains 62 publications from 24 countries. Figure 5 illustrates the distribution of the publications from various parts of the world. In general, 16.1% of publications were executed in United States and Italy each (10 articles each), 8.1% were reported from Netherlands (5 articles). 6.5% were published from India, United Kingdom, and China each. Another 5% were published from South Africa (3 articles). In addition, 3.2% were reported from Romania, Korea, Germany, Canada, and Switzerland each (2 articles). Taiwan, United Arab Emirates, Portugal, Spain, Indonesia, Czech Republic, Singapore, Brazil, Finland, Turkey, Bangladesh, and Ireland have the share of 1.6% each, from the whole 62 reviewed publications.

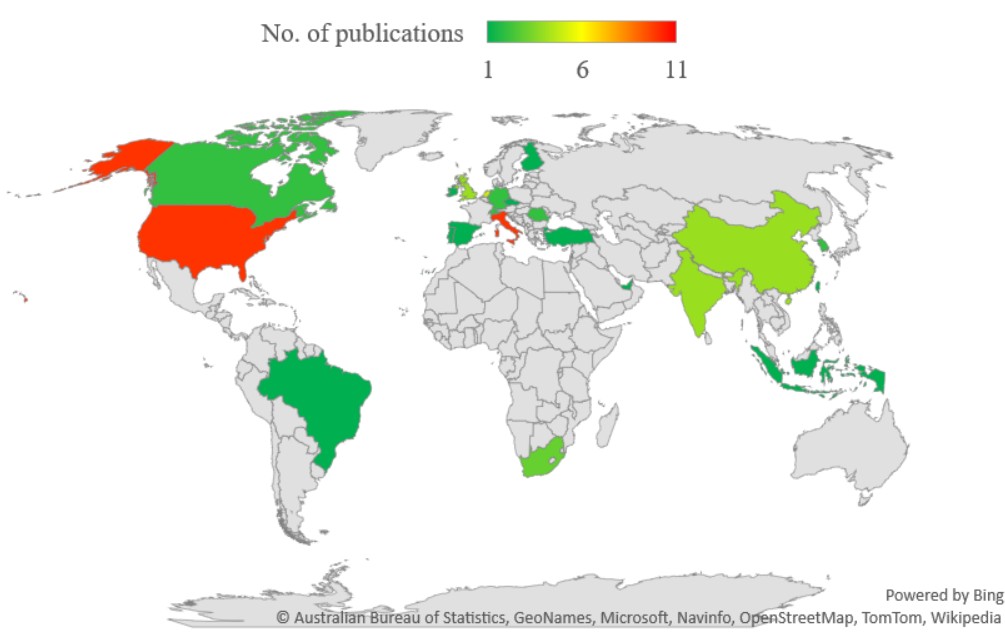

**Figure 5.** Distribution of the articles associated with indoor monitoring of buildings using low-cost sensors.

3.1.1. Energy Efficiency of the Electrical Appliance (Category 1)

This sub-section provides detailed information of the reviewed articles connected with the category 1, low-cost monitoring of the electrical energy consumption in buildings. Electricity is one of the fields of energy that has higher range of the annual statistical reports (in the news) in the world than the other sources of energy such as coal, natural gas, oil, and renewables [103]. According to the reports, since 1974 the global electricity consumption has raised year by year continuously except 2008 and 2009 when worldwide economic crisis occurred [102]. Electrical energy efficiency in buildings can be defined as diminution in the power consumption of electrical devices/appliance without changing normal life of the habitants. In fact, one of the most common way in the literature to adjust the energy consumption, is visualization of occupancy information through different technologies, such as: webcam, microphone, infrared-red sensor, proximity sensor, air temperature and humidity sensor, light sensor, passive infrared sensor, occupancy sensor, $CO_2$ sensor, and acoustic sensor. For instance, Cheng et al. [104], Longo et al. [105], Pereira et al. [106], Monti et al. [107], Labeodan [108], Foster et al. [109], Zhao et al. [110], Lan et al. [111], Paci et al. [112], and Chen and Ahn [113], deployed occupancy sensors in the room to detect the people presence to reduce the energy consumption. Vanus et al. [114] installed a $CO_2$ detection sensor to obtain information on the occupancy of the building. Dedesko et al. [115] presented dual-sensor methodology ($CO_2$ concentration sensor and non-directional doorway beam-break sensor) to provide a low-cost and accurate estimation of occupancy in the buildings. For detecting the indoor activities, Nguyen and Aiello utilized a low-cost acoustic sensor [116]. Wang et al. used dynamic spatial occupancy distribution (DSOD) as a new technique to infer occupancy distribution through low-cost Bluetooth energy network. Newsham et al. mounted a low-cost webcam and microphone to present an accurate indoor occupancy sensing [117]. For optimal estimation of occupancy based on the data of the light LCSs, Chen et al. used Bayes filter with neural network [118]. Chaney et al. used the data extracted from temperature and $CO_2$ sensors to estimate building occupancy [119], controlling the house appliance for the energy efficiency of the building through low-cost internet protocol presented in [63].

### 3.1.2. Controlling of the Air Quality and Pollution (Category 2)

The four principal indoor air pollutants are asbestos, radon and carbon dioxide. However, the whole reviewed publications in this area were focused only on low-cost detection of the aerosol and $CO_2$ in the buildings. In this category (category 2), thirteen of the reviewed articles were focusing on the indoor air pollution to improve the overall health of the occupants. According to the field test, the reviewed articles in this category can be classified to three groups. (1) a group of articles implementing the monitoring devices in different types of buildings such as educational, historical, residential and laboratory. (2) a group of articles that used laboratory prototype for calibration of their monitoring system. (3) a group of articles that did not explicitly determine the case studies but reported the methodology for development of the low-cost $CO_2$ sensing devices. Figure 6 provides information about percentages distribution of the 3 groups of articles dealing with controlling of the air quality and pollutions in buildings.

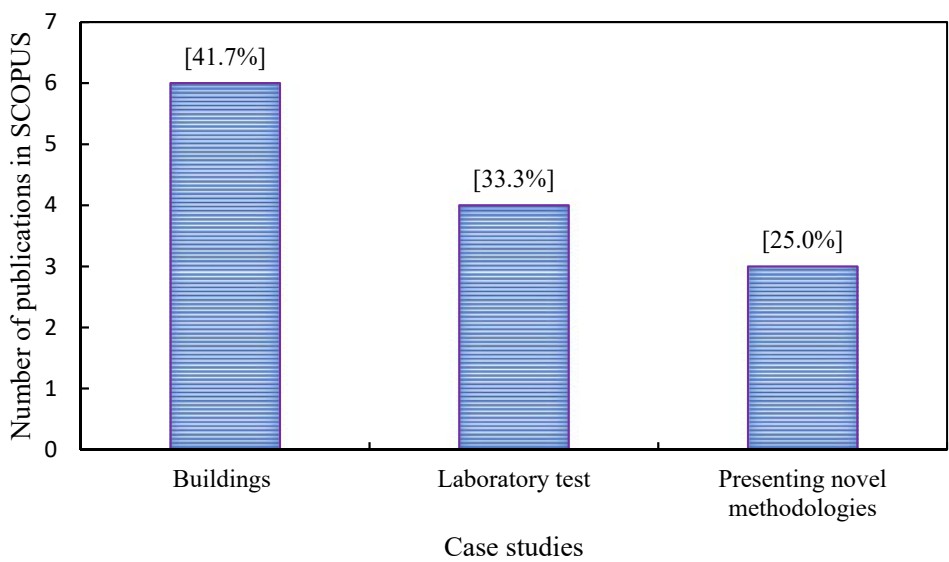

**Figure 6.** Classification of the articles associated with controlling of the air quality and the pollutions inside buildings. Group 1: focusing on building sector, group 2: conducting laboratory experiments, and group 3: presenting novel methodologies.

It is worthwhile to note that, 41.7% of the investigated articles, carried out field assessment of indoor air quality in educational buildings [120], historical buildings [121], residential buildings [122,123], and laboratory building [124]. One-third of the authors used laboratory prototype for calibration of the sensors. For instance, Lachhab et al. proposed a wireless sensor network for indoor and outdoor monitoring of $CO_2$. To do so, they developed a prototype in the laboratory and controlled the function of the ventilation system [125]. Herrick et al. presented an initial evaluation of the $CO_2$ sensor in a testing chamber [126]. Examples of assessing precision of the low-cost particle sensors based on the light scattering in a glass chamber can be found in [65,127]. The remaining publications in the group 3 did not clearly state their case study as they dedicated their study to describe the development of the low-cost $CO_2$ sensing system based on the nanofibers [128], wireless sensor [129] and hot-wire $CO_2$ sensor in CMOS technology [130].

### 3.1.3. Thermal Comfort and HVAC (Category 3)

In the third category, a total of 24 articles were analyzed as they carried out assessment of the thermal comfort and satisfaction of the buildings' habitants by measuring the indoor temperature and humidity in their studies. There are three different approaches in the literature regarding HVAC and thermal comfort monitoring, which are: (1) a group of authors focused only on the public buildings, (2) another group of authors that focused only on

the residential buildings, and (3) a group of authors presenting low-cost methodologies to determine thermal comfort through extracting the skin temperature of occupants. Figure 7 Provides information about percentages distribution of the said 3 groups of articles dealing with thermal comfort and HVAC.

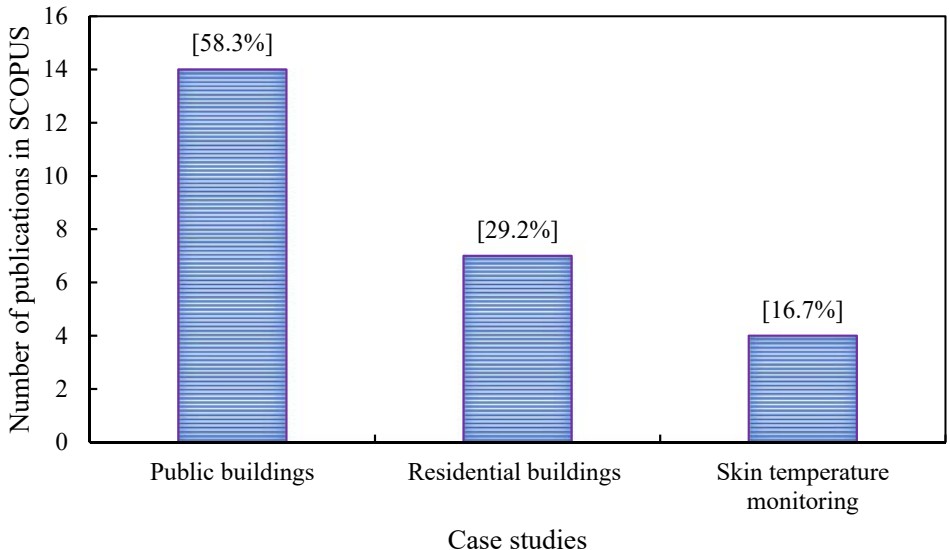

**Figure 7.** Three groups of case studies in the SCOPUS database for application of low-cost sensors on thermal comfort monitoring and HVAC in the building sector. Group 1: focusing on public buildings, group 2: focusing on residential buildings, and group 3: focusing on measuring skin temperature of people.

The first group of authors were deploying the LCSs in the public buildings for assessing the satisfaction level of the people in terms of thermal comfort. This group of the articles contains 58.3% of the publications dealing with thermal comfort and HVAC monitoring (e.g., controlling temperature and humidity). Examples of the public places can be referred to: (A) Educational places. Viani and Polo installed a cheap wireless HVAC system in a research center in University of Trento in Italy for monitoring indoor situation of and managing the energy system [131]. Chen and Chan focused their studies on the thermal comfort assessment of students in the humid climate of Taiwan [132]. Furthermore, applicability of the low-cost HVAC systems in densely crowded places, such as schools, were checked by Perez et al. [120]. Hossain et al. presented implementation of IoT for the high density of environmental monitoring. In this educational workshop, sixty LCSs were installed in the three floors of the University of Westminster in London [98]. (B) Heritages and historical places. Low-cost strategies for preventative protection of the cultural places through long-term scattered monitoring of humidity and temperature can be found in the publication of D'Orazio et al. [133]. The other groups of authors were focusing on the other public places, such as mosques [134], commercial offices [135], and laboratories [136]. Barrios and Kleiminger presented a low-cost data acquisition system to enhance the efficiency of heating and cooling systems in the buildings [137]. They also studied the effect of thermal comfort on the metabolic rate of people. Kumar et al. dedicated their study to present the thermal comfort index for buildings. To do so, they developed a low-cost HVAC system and compared the obtained results with the subjective responses and predicted mean vote (PMV) value [138].

As shown in the Figure 7, the second group of studying low-cost thermal comfort monitoring, the authors focused their research on the monitoring of the residential buildings. In fact, providing comfort level of the habitants is one of the main aspects in residential building automation. However, real time monitoring of HVAC is expensive and difficult. This approach contains 29.2% of the reviewed articles in terms of thermal comfort monitoring of the buildings' habitants. Yang et al. developed a low-cost HVAC system to

control a residential building automation. In this study, various sensor nodes including: temperature, humidity, air quality, RFID reader, illumination level, and fire level in each room of the building [139]. Spatial distribution of the low-cost sensor nodes associated with thermal comfort monitoring (humidity, temperature, $CO_2$ detection) was carried out by Vasilievici and Costea [140]. Li et al. developed the multi-zone HVAC controlling system in a residential house. In fact, another focus of the authors in this investigation was to study the factors that affect wireless data transmission in residential places, such as walls, floors, and furniture [141,142]. Kumar et al. presented a low-cost comfort sensing network considering reliability, portability, and the consumption rate [143]. In this study, calibration of the system was carried out by comparing the observations of the humidity sensors with an EM5510 multimeter. Taking into account provision of the IEE1451 standard, the presented real-time monitoring system was proposed for monitoring of homes, hospitals, and automobiles.

The third and, also, last approach of thermal comfort monitoring is based on measuring skin temperature of the people (as shown in the Figure 7). Various types of infrared devices reported in the literature, tested in several situations to perceive people comfort level. Through body temperature monitoring, Wu et al. used a low-cost thermal camera (which uses low-cost infrared sensor MLX90640), to assess people thermal comfort [144]. This low-cost camera has resolution of $32 \times 24$ pixel. The accuracy of the device for measuring environmental and skin temperature are 0.1 °C and 0.15 °C, respectively. A new paradigm as human embodied autonomous thermostat introduced by Li et al., which was based on the facial skin temperature scanning to maintain the people comfort level and attenuate the use of energy in the buildings [145]. This technique can save the energy consumption of buildings up to 30%. For the same purpose, FLIR Lepton low-cost thermal camera with accuracy of 2 °C and resolution of 0.1 °C and RGB camera with was used by Aryal et al. to extract people skin temperature [146]. Real time monitoring of skin temperature through FLIR thermal camera S65-HS reported by Vissers and Zeiler [147].

### 3.1.4. Other Aspects of Indoor Monitoring of Buildings (Category 4)

In some few instances, the studies followed a slightly different approach than the last three groups, so the authors decided to include them within a fourth group: group 4. In this sense for example, a wireless methodology for low-cost installation of chemical sensors to detect whether chemical events is occurred in the building can be find in [148]. Taking the advantage of straw bales and their installation in the wall of the buildings, Lawrence et al. monitored fluctuation of the moisture level up to ten months after the sensors' installation [149]. For the same purpose, Davies and Ye developed pad sensor to accurately measure the moisture level of the structures' envelopes by simply attaching the pad sensor to the element [150]. Examples of a low-cost pressure monitoring system for identifying the water consumption activities in buildings from a single installation point can be found in [151,152]. The low-cost energy management of package rooftop air conditioner presented by Yu et al. [153]. Long term passive environmental monitoring of heritage buildings in terms of climate variations was proposed Balsamo et al. [154]. Example of Arduino based low-cost fire detection system also reported in [155]. Low-cost thermal parameters diagnosis of the building envelope based on the wireless sensor network reported in [156]. Andreottei et al. developed a novel hot box to derive thermal parameters of masonry and historic buildings, such as heat flux, surface temperature, relative humidity, and air temperature [157]. The main feature of the developed metering device are: reliability of the monitoring system, conservation of cultural heritage, ease of equipment installation, and, finally, economic cost [158].

### 3.2. Structural Building Monitoring

In this section, low-cost approaches for monitoring structural parameters of buildings are overviewed. Structural monitoring of the buildings using LCSs is the second group of work analyzed. Controlling the aging process of buildings by monitoring acceleration, as well as strain or displacement is crucial for structural safety. Therefore, in this section the existing low-cost technologies for the measurements of the acceleration and displacement or strain are reviewed. Figure 8 illustrates the increment of publications over the years associated with low-cost structural building monitoring. In fact, in the two main subcategories of structural health monitoring, detecting structural response through determination of vibration pattern in buildings is more decisive as it contains 56.8% of reviewed articles published until 2020. The articles of this part were limited to the publications from 2006 onward as this is coincident with development of Arduino at the end of 2005. The first article founded in 2006 aimed to reduce the labor and cost associated with building monitoring [159]. They used low-cost analog sensors and ATmega 128 microcontroller for real time structural monitoring of a half-scale building installed on a shaking table. It can be seen that the quantity of the annually publications related to the low-cost accelerometer are so limited from 2006 until 2013 as they have the share of 28.5% in the whole reviewed articles. However, the number of papers published in the last 7 years has increased by 150% and has the share of 71.5%. Such an increment is in line with the developments of the low-cost data acquisition systems using microcontrollers and low-cost kits.

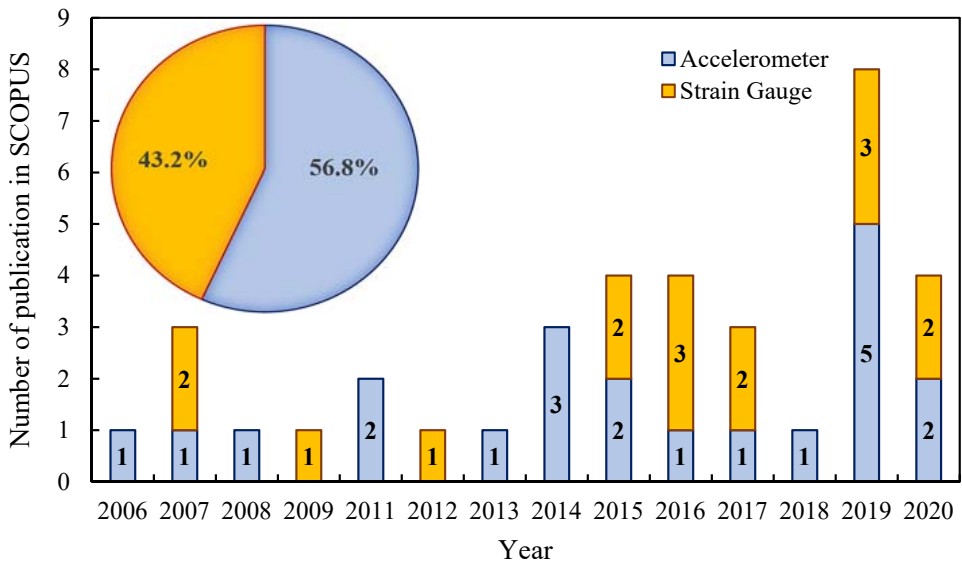

**Figure 8.** Distribution of the reviewed articles associated with the application of LCSs for structural monitoring from 2006 to 2020.

Figure 9 illustrates the number of articles by country. It can be observed that Italy and USA had the most contributed to the low-cost structural building monitoring by reporting 12 and 8 publications, respectively. Then, 3 publications were carried out in China and Taiwan each. Additionally, 2 articles were executed in Japan, and the quantity of the publication for the other 10 countries in the map (Serbia, Portugal, Spain, Indonesia, India, Australia, Greece, Korea, Poland, and Germany) were 1 each.

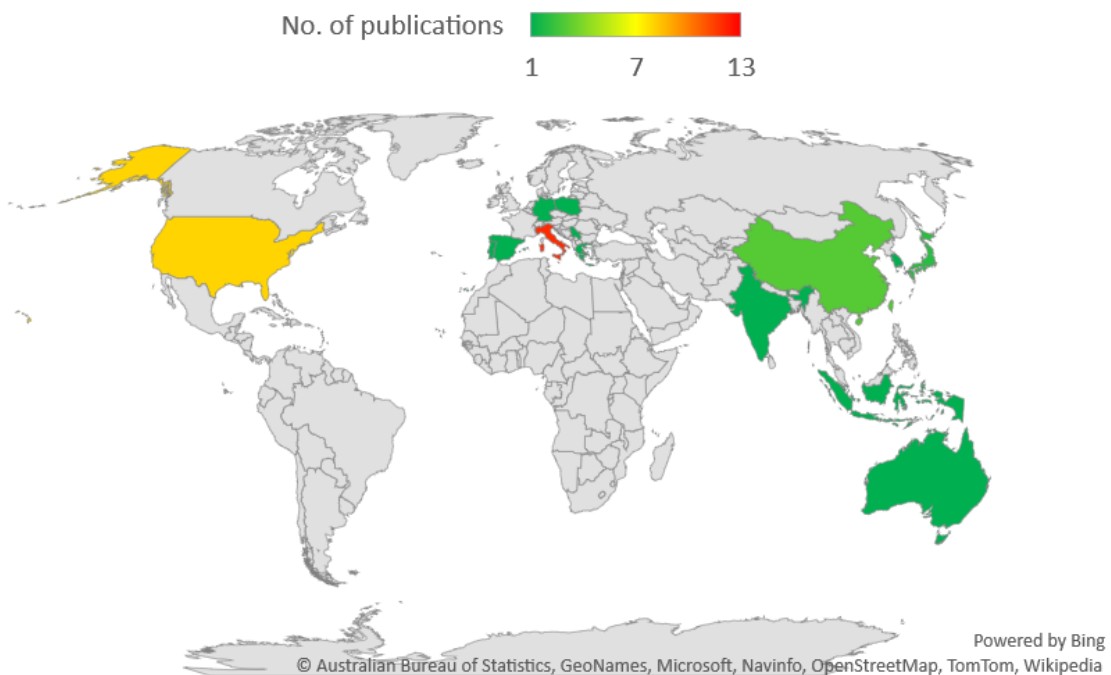

**Figure 9.** Distribution of the articles associated with low-cost structural building monitoring.

### 3.2.1. Vibration Monitoring (Category 5)

In this subsection, the articles that were focusing on the low-cost approaches for acceleration measurements in the buildings are reviewed. After implementation of the filtering process (presented in the Figure 2), 21 articles were assigned to this section. As shown in the Figure 10, three groups of publications found in SCOPUS used low-cost accelerometer in their research. These three groups of authors focused their research on monuments and old buildings, public and residential buildings, and finally scaled building models. In the first group, only two articles recently (2019 and 2020) utilized low-cost accelerometer for low-cost structural monitoring of historical buildings, one in Greece [160] and the other one in Italy [161]. Afterwards, the second group of scholars focused their studies on implementing the devices on residential and public buildings [162–164]. For instance, Pierleono, et al. presented an IoT solution based on the message queue telemetry transport protocol (MQTT) for real time structural monitoring of the buildings [165]. In the other article, the authors designed a wireless sensor network system for real-time monitoring of acceleration and also environmental parameters at the same time [166]. Application of the fiber optical sensor (FBG) based accelerometer for structural monitoring of the buildings can be found in [167]. These types of sensors have the benefits of low transmission losses, light weight, electrical isolation, and immunity to electrical interference. Thus, it can be proposed to readers for future structural monitoring projects. Kohler et al. installed cheap seismometers in building in Los Angeles to assess shaking intensity caused by earthquakes [168]. In fact, the major objectives of the developed protocol were to compute the modal characteristics of the structure such as mode shape and frequencies. Due to the high level of earthquake activities in New Zealand, Simkin et al. instrumented a couple of houses in Wellington with inexpensive accelerometers to monitor the dynamic behavior of the buildings during the earthquake excitations [169]. Integration of the low-cost structural health monitoring with the building management system (Gas and electricity) can also be found in [170].

The third category of the publications associated with low-cost development of acceleration in buildings contains 9 articles. Various methods have been proposed by different authors for evaluating the dynamic behavior of the scale structures in the laboratory [159,171–173]. A wireless low-cost data acquisition system implemented by Lynch et al.

and mounted on a three-story building for real time monitoring of acceleration and velocity of different degree of freedom of the case study [173]. Another example of wired/wireless monitoring system installed on a scale building in the laboratory reviewed in [174]. Hosseinzadeh and Harvey developed a cost-efficient technique for the scale building monitoring through several surveillance cameras [175]. Ando et al. proposed a low-cost wireless sensor networks using multiple low-cost triaxial accelerometers and inclinometers for structural building monitoring [99]. Application of low-cost MEMS seismometer can be checked in [176].

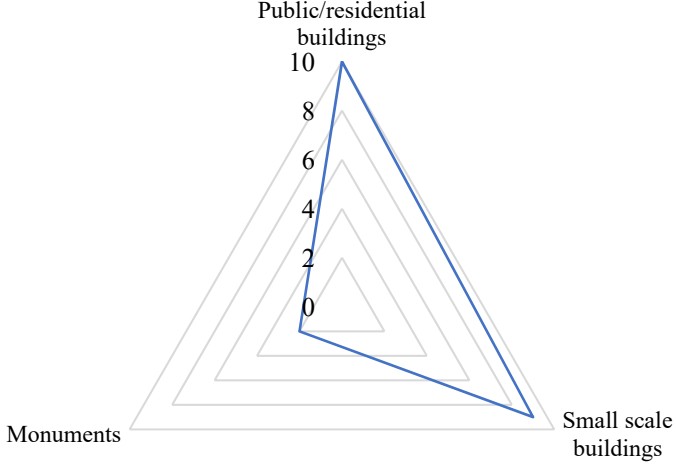

**Figure 10.** Number of studies carried out using different approaches for low-cost acceleration monitoring in buildings.

### 3.2.2. Strain Monitoring (Category 6)

The rest 43.2% of publications remains from the structural health monitoring, discussed implementation of low-cost sensors for monitoring of structural deformation, damage detection and the measurements of strain and displacement at various parts of the buildings. The first publication associated with this area dated in 2007. In this publication Tosi et al. focused their studies on a cheap interrogation system of fiber Bragg gratin sensors for crack monitoring and measuring the strain up to 320 $\mu\varepsilon$ with precision of 3% [177].

As demonstrated in the Figure 11, all the founded articles dealing with SHM of the buildings in terms of damage and displacement detection, are grouped into three categories. The first group of the authors strictly focused on the monuments. The three reviewed articles in this group provided insight into inexpensive monitoring of ancient city wall [178], old church [179], and old court [180]. In the second group of the publications five authors presented implementation of their sensors by testing them on the scale beam [181], building model in the laboratories [182,183], and measurement of the shaking table by smartphone [184]. At the smaller scale, publications of the third group focused on the experiments on the laboratory specimens such as: Liu et al. [185], Zhang et al. [186], Sasaki et al. [187], Heyse et al. [188], Liu et al. [189], and Shiraishi et al. [190]. For instance, Olivero et al. conducted an extensive test in an environmental chamber to present a novel low-cost optical data acquisition system for monitor the evolution of cracks [191].

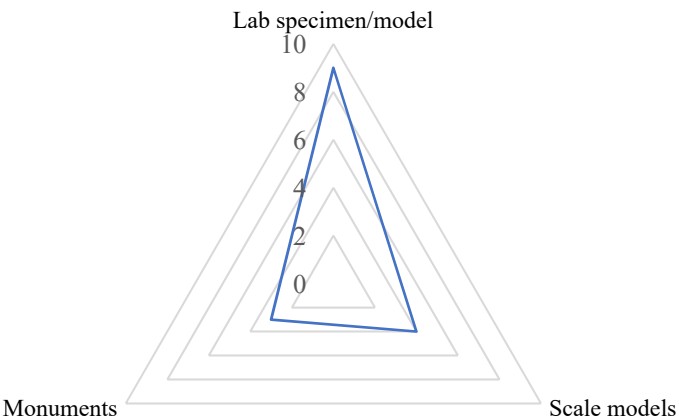

**Figure 11.** Number of studies carried out using different approaches for low-cost displacement monitoring in buildings.

## 4. Studying Microcontrollers and Communication Protocols in Literature

This section presents types of microcontrollers utilized for low-cost monitoring projects. In fact, the answer to question 1 (What are the preferred types of microcontrollers for low-cost monitoring?) presented in the Section 2.1 of the article (Research question) provides insight into modularity and flexibility of the most popular microcontroller for the field of building monitoring. Full-fledged bank-card sized computers are the core of the majority of the reviewed papers presenting low-cost data acquisition system. Examples of these microcontrollers are Arduino Uno [121], Arduino MEGA [136], Arduino Due [192], Arduino Nano [127], Raspberry Pi [137], Orange Pi [193], STM32F7 [194], ESP8266 [195], Thikerforge Bricklet [98], GR-SAKURA [174], Beaglebone [196], PIC16F873 [162], PIC18F4620 [197], PIC18F458 [141], and PIC18F45K50 [188]. Considering the aim of the metering system to be established, the options provided by the microcontroller, nature of the experiment, and the processing capacity of the microcontroller, a user can choose the most adequate microcontroller. For instance, Wang et al. stablished a compact weather station by means of Arduino UNO microcontroller [198], Nayyar and Puri gave comprehensive examples of Beaglebone technology for the aim of IoT and various aspects of the building monitoring [196], and Imteaj et al. employed Raspberry Pi to design of fire alarm system [155]. Kusriyanto et al. conducted a monitoring system based on the Arduino MEGA to allow the users to monitor and control the house appliances [63]. Example of low-cost image acquisition system built using STM32F7 microcontroller can be found in [199]. Chase et al. developed a solar irradiance monitoring system installed on ESP8266 microcontroller [70]. Aiming at developing a low-cost structural monitoring system, Liang et al. installed MEMS sensor on a single-board GR-SAKURA microcontroller to measure acceleration parameter [174].

Analysis of Table 4 shows distribution of the microcontrollers utilized for the developments of the low-cost building monitoring systems in the SCOPUS database. It can be seen that 17 types of microcontrollers were used for this aim and among them Arduino UNO (9), Arduino Mega (9), and Raspberry Pi (7) were the most commonly used between scholars. The remaining 12 microcontrollers were utilized in a limited number of publications. The differences between Arduino and Raspberry Pi are obvious and, vast as they are, are called microprocessor and mini operational computer, respectively. In fact, choosing between these two families of microcontrollers is a question of the project requirements. Raspberry Pi is the best choice when a user requires a full-fledge computer for conducting various tasks and heavy calculations at a same time and also in real-time. This is due to the higher speed (up to 40 time in the clock speed) and RAM (around 128,000 time) of Raspberry Pi than Arduino. Moreover, Raspberry Pi provides a camera port, 4 USB ports, HDMI port, LCD port, micro-UBS port, and 1 DSI display port, which makes it appropriate for multiple applications. Arduino does not have many of the mentioned options unless the user adds them through shields. However, Arduino has many advantages over Raspberry Pi which

made the researchers favor of it, such as: (1) Simplicity: it is so simple to interface digital and analogue sensors and other electronic elements with Arduino. (2) Price: all the models of Arduino are much cheaper than those of the Raspberry Pi. (3) Power consumption: Arduino runs on very low power and depends on monitoring duration, a required power can be supplied by power bank. However, Raspberry Pi requires continuous 5 V power supply which it is so thought to supply it using power bank. (4) Robustness: As Raspberry Pi runs on OS, power cut causes damage to the software and applications, thus it must be shut down before the power cut off. However, with Arduino, if power disconnecting occurs it again restarts and it keeps working properly.

**Table 4.** List of the microcontrollers used in SCOPUS database for low-cost building monitoring.

| Microcontroller | Number of Publications | References |
| --- | --- | --- |
| Arduino UNO | 9 | [65,99,104,121,125,129,137,178,200] |
| Arduino MEGA | 9 | [63,136,143,156,159,160,173,178,201] |
| Raspberry Pi | 7 | [125,132,134,137,155,161,202] |
| Arduino Nano | 3 | [109,127,203] |
| ADS8344 ADC and AVR | 2 | [151,152] |
| STM32F303 | 2 | [165,172] |
| MSP430 | 1 | [116] |
| PSoC | 1 | [120] |
| PIC16F873 | 1 | [162] |
| PIC18F4620 | 1 | [197] |
| PIC18F458 | 1 | [141] |
| PIC18F45K50 | 1 | [188] |
| Tinkerforge Bricklet | 1 | [98] |
| GR-SAKURA | 1 | [174] |
| AT90S8515 AVR | 1 | [163] |
| TI MSP430 | 1 | [166] |
| NXP JN5148 | 1 | [154] |

Due to an increasing attention to low-cost building monitoring, demands for secure communication between the network nodes have been increased. LCSs are deployed on a large number of nodes. Therefore, there are multiple secure and authentic communication protocols developed by companies for self-organization of monitoring facilities. To select a transmission data protocol, it is imperative to determine its specific operation and the network needs. To answer question 2 (What are the chosen communication protocols?), Table 5 is presented, as it provides the preferred data acquisition methods in the reviewed papers. The obtained results show that ZigBee radio frequency (RF) protocol addresses the requirements of low-cost building monitoring systems more than the others. In fact, between the reviewed articles, 18 monitoring systems were stablished based on this technology. This can be explained as: (1) It offers flexible network structure: This protocol offers a simple, intelligent wireless data transfer solution which is regularly being used for building automation and smart communication. (2) It provides long lasting battery life: it can work for years on cheap batteries for a host of diverse monitoring projects. (3) Is a wireless mesh network: unlike Bluetooth which runs point to point communication, Zigbee provides mesh peer to peer network. This implies that data transmission accomplishes through multiple hops and if transmission fails, the node discovers the other route to arrive to the defined place. This feature also makes possible to have an expandable network up to 65,000 nodes across a vast area on a monitoring system. Depends on the XBee module

being used, it offers communication range around 100 m in closed places and 1000 m in open area. However, it has some deficiencies such as low data transmission rate (20 Kbps, 40 Kbps, and 250 Kbps), less security than the other protocols same as Wi-Fi based communication system and Bluetooth. Wi-Fi was the second preferred communication protocol in the reviewed articles and 10 monitoring systems were controlled by this protocol. This technology provides wireless communication between different modules of monitoring system using Internet connection. It uses radio frequency for sending signal between devices. It is possible to control the bandwidth usage and it has high transmission speed (for a moderate protocol up to 600 Mbps). However, the limitation of this protocol is the high-power consumption. The third preferred communication protocol was radio frequency as 7 research project carried out based on this protocol. Only 2 articles were using SD card for saving the data. The rest of the protocols such as Bluetooth, Ethernet, direct wire connection to logger, and MQTT were used in a single article each.

**Table 5.** Group of communication protocols was used in SCOPUS database for low-cost building monitoring.

| Communication Protocol | Number of Publications | References |
| --- | --- | --- |
| ZigBee | 18 | [99,104,108,109,121,127,129,131,139–142,156,160,166,174,178] |
| Wi-Fi | 10 | [98,106,110,112,113,117,132,136,137,202] |
| Other Radio Frequency | 7 | [125,159,162,163,183,189,203] |
| SD-card | 2 | [143,156] |
| Bluetooth | 1 | [202] |
| Ethernet | 1 | [188] |
| Wired connection | 1 | [172] |
| MQTT | 1 | [165] |

## 5. Integration of BIM and Low-Cost Sensors

This section investigates examples of articles dealing with integration of LCSs and BIM. The first concept of BIM was proposed in a book entitled "Building Product Models" written by Eastman in 1993 [204]. This book introduces the principal idea of architectural information modeling, linked model components, and information changes. The growing interest in integration of LCSs and BIM in SCOPUS database is illustrated in Figure 12. It shows that connection of these two technologies is significantly growing with time. For example, this is indicated by the number of total publications on this topic, which it has grown from 1 in 2011 to 58 in 2020.

There are diverse research lines in the literature dealing with application of BIM to low-cost monitoring of buildings. Some of the authors focused on the connection of low-cost data acquisition system to BIM model [205]. In fact, wise selection of the methodology for transferring the data from sensors to BIM model may help the users to make real-time monitoring of the measurements, accurate inclusion of geometric shapes and associated data, and finally avoiding the loss-data. Kensek investigated feasibility of connecting environmental sensors, such as light, humidity, and $CO_2$ to BIM. To do so, they have tried to make the Arduino–Revit connection through Dynamo, Rhino, and Grasshopper [206]. Chang et al. described a methodology for visualizing measurements recorded with sensors in BIM models, employing several perspectives in which to support complex decisions that require interdisciplinary information. They utilized Dynamo to bring procedural information from Arduino into 3D BIM. Moreover, they focused on the design of a platform for the connection of low-cost sensors to various microcontrollers (Arduino and Raspberry Pi) [207]. Gunduz et al. presented a sample software architecture for integration of BIM, geographical information system (GIS), and IoT for supporting comfort analysis [208]. The

authors used Arduino UNO to collect measurements from LCSs of temperature, humidity, and light level. Shahinmoghadam et al. studied the synergistic benefits of BIM, IoT, and virtual reality (VR) for establishing a low-cost monitoring thermal comfort system [209].

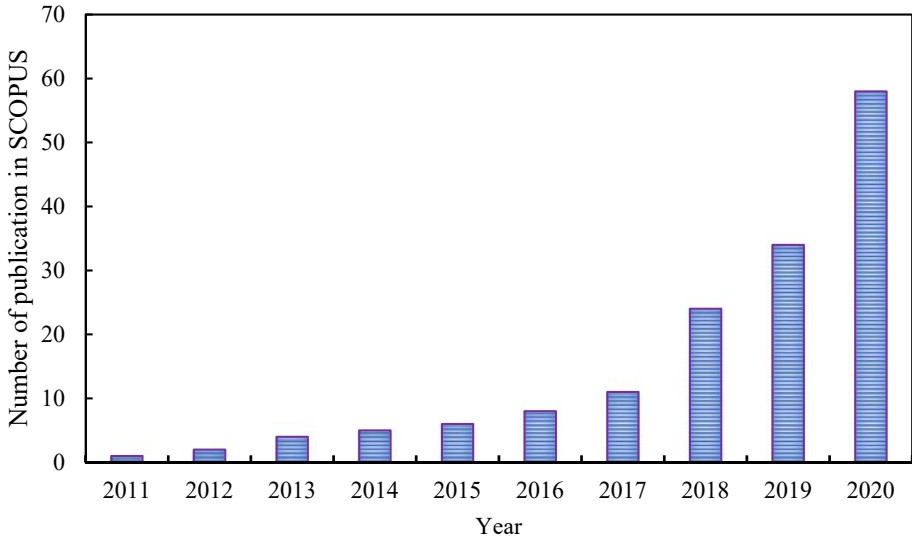

**Figure 12.** Number of works addressing integration of BIM and LCS in the Scopus database from 2011 to 2020.

However, the other group of authors have been mainly focused on integration of BIM and LCSs to support visualization of live data streams for monitoring various parameters of buildings [210]. An Arduino-based data acquisition system developed by Natephra and Motamedi for real-time monitoring of thermal comfort parameters, such as: indoor temperature, humidity, and light intensity [211]. Desogus et al. developed a BIM platform for real-time monitoring of the data (temperature and luminance) coming from low-cost sensors within parametric model of a historical building [212]. Li et al. proposed an automatic as-built BIMs framework that transforms the noisy 3D point cloud generated by low-cost RGB-D sensor [213]. Shen and Lu proposed a new methodology to engage the BIM model as a control system of building energy performance. They designed a parametric adaptive skin system (PASS) to combine the adaption of natural sunlight use for higher building performance [214]. Rahmani Asl et al. developed a framework for BIM-based performance optimization. They used BPOpt in reducing the energy consumption while increasing the sufficient daylight level for buildings [215]. Teizer et al. focused on permanent availability of actual performance datasets through IoT, which combines environmental information in a cloud-based BIM platform [216].

## 6. Discussion

In fact, answering the first and second questions was carried out in the previous section (Section 4), as they were in line with the architecture of the monitoring devices and also reviewing of the microcontrollers and communication protocols in the literature. To answer question 3 (What are the most important parameters to be monitored in the field of building monitoring?), it has to be mentioned that according to the presented results the most desired parameters to be monitored in the building sector were temperature and humidity. This is due to the fact that in the current century people are spending 90% of life inside buildings, which necessitates providing thermal comfort condition for the habitants [190]. In fact, the economic motivations for spending money in the building sector to raise productivity of working staff is unquestionable [217]. It was reported that attenuation of working performance by 0.5%–5% causes a loss of USD 20 to 200 billion per year in the United States [218]. Multiple low-cost thermal comfort monitoring systems were presented in the reviewed publications for real-time wireless monitoring applications [98,131,134,136],

that can be good references for readers. However, before using the proposed references and making the final decision for selection of the components of a monitoring system, engineers must take into account the dimension and requirements of monitoring site. These preliminary considerations contain the capability of the microcontrollers, the detection range of sensors and physical range of communication protocols. Wise selection of said elements affects accuracy of the monitoring and also increases the decision-making for habitant's health condition. For instance, the most common low-cost sensors used by the authors for the indoor temperature and humidity monitoring are SHT-31D [133], DHT22 [202], HH4000 [138], LM35CAZ [138], and BME280 [200]. The main concerns of the wireless monitoring of the indoor parameters in public buildings are: [1] power supply and [2] the distance between sensor nodes and the data logger. To deal with problem of power supply, majority of the reviewed papers were using the Zigbee communication protocol to design a power-efficient monitoring system. For solving this issue, solar cells can be a reliable alternative for low-cost monitoring projects. Parallel to this technology, consideration of a power bank at each measurement point can alleviate the concern of power-cut in cloudy days.

The least number of publications in the group of indoor monitoring of buildings were carried out for monitoring the indoor air quality through the measurement of the $CO_2$ and pollutant emission. There are principal elements connected with people health condition and working performance and the indoor air quality. These elements are infectious illness, respiration problems, asthma, and sick building syndrome symptoms. It was appraised that in United States the annual saving of USD 6 to 19 billion comes from reduction in respiratory disease, USD 1 to 4 billion from reduction in allergies and asthma, and USD 10 to 20 billion from reduction in syndrome symptoms [218]. Overall, 95% of the population in low and median salary countries consume solid fuels for their cooking and heating requirements, which can be a source producing $CO_2$ in the buildings. According to environmental protection agency, indoor air pollution is also influencing by the other sources, such as cleaning products, carpet, and painting. Consequently, the impact of indoor air pollution is almost 100 times higher than the outdoor level of pollution [81]. In fact, in future more focus must be given to presentation of low-cost air quality monitoring systems for poor ventilation system of the residential buildings, as well as public buildings such as hospital, school, and the other public buildings. Future studies on the school sector must concentrate on monitoring of $O_2$, VOCs, HCHO, $CO_2$, fungi, and bacteria [219,220]. In the case of hospitals, the attention should be paid on monitoring of $CO_2$, VOCs, CO, humidity, and temperature [221]. In the case of administrative buildings, the focuses should be given to monitoring of $CO_2$, CO, PM, VOCs, humidity, and temperature [222]. Finally, for the houses the essential parameters to be monitored are radon, PM level, $CO_2$, humidity, and temperature.

It is noteworthy that the waste of energy is directly proportional to the age and the quality of insulation of the buildings. Therefore, thermal parameter diagnosis of the building envelope helps engineers to estimate the energy performance of the building and reduce energy consumption. In the reviewed articles, a few groups of authors focused on presentation of low-cost thermal monitoring of buildings. These parameters are for example transmittance and resistance values that represent thermal performance of buildings [156,157]. Therefore, it is of paramount importance to integrate monitoring systems with low-cost ambient and surface temperature sensor to provide high density of measurement points for acceptable assessment of buildings' thermal performance.

A multiple methodology for damage detection, vibration controlling, model updating, and safety assessment of buildings reviewed in the cited articles. Different low-cost sensing systems have been proposed from 2007 to 2020 for damage detection of small scaled and real scaled buildings. Among them, fiber optic sensors present substantial prospects for damage detection of structures. In the literature, fiber optic technology has been broadly utilized for short-term monitoring of building elements in laboratories [151,177,181]. Whole

applications of wireless monitoring systems have proven the potential to be applied for vibration measurement of real size buildings or small scaled size in the laboratory.

## 7. Conclusions

In this article, a systematic literature review of application of low-cost sensors for building monitoring has been presented. After implementation of inclusion and exclusion process of the founded articles in SCOPUS database, 99 linked articles from 2006 to 2020 were chosen and discussed. According to the reviewed articles, we could provide the readers an overview of the parameters to be monitored in building sector, including indoor field (electricity consumption, air quality, and thermal comfort and HVAC) and structural field (vibration and strain). In the case of assessing the thermal comfort and satisfaction of buildings' habitants, it was found out that majority of the authors (58.3%) focused their studies on monitoring of said parameters in public buildings. This is due to the fact that, the people spend 90% of the life inside buildings. In addition, the economic motivations for spending money in building sector to raise productivity of working staff is unquestionable. As it has proven that attenuation of working performance by 0.5%–5% causes a loss of USD 20 to 200 billion per year. The reviewing of the articles for low-cost monitoring of the indoor air quality indicated that 41.7% of scholars were focusing on implementation of their devices on monitoring of various types of buildings, such as educational, historical, residential, and laboratory. Whereas the annual saving of USD 6 to 19 billion comes from reduction in respiratory disease. Therefore, future studies should focus on developing low-cost monitoring systems for detection of $O_2$, VOCs, HCHO, $CO_2$, fungi, and bacteria in schools. In the case of hospital researchers should focus on measuring the level of $CO_2$, VOCs, and CO. In the case of administrative buildings future studies must be carried out for monitoring of $CO_2$, CO, PM, and VOCs. In the case of residential buildings, it is also essential to focus on monitoring the PM and $CO_2$ level.

Regarding the architecture of the low-cost data acquisitions systems, two tables providing list of utilized microcontrollers (17 types) and communication protocols (8 types) in the literature were drawn. It has been discovered that Arduino microcontroller and ZigBee communication protocol have attracted a great deal of attention. It is important to highlight that when deploying low-cost devices for building monitoring, it is essential to determine the influential factors, and also enhance the precision of monitoring while holding the simplicity of the operational principals. When it comes appraising the methodologies for data processing of the low-cost monitoring systems in the literature, it has been found that majority of the authors have used open-source models that are peer production and encourage open collaboration. This approach led to massively reduction in the monitoring costs in the literature, as the source codes, libraries, software are freely available for feasible modifications and redevelopment by any users. In addition, integration of low-cost monitoring devices and BIM model has been studied and growing interests on this field has been analyzed. This article reviewed several attempts for designing of platforms to receive raw sensors data as an input and delivers automatic visualization of the processed data in BIM. In fact, the most common platform in the literature for connection of microcontrollers and BIM model is Dynamo. Reviewing of the other articles indicated real-time streaming of the sensor data is not limited only to BIM technology, as IoT and VR has been implemented in various low-cost monitoring projects to increase the project productivity.

This article indicates that although remarkable number of studies has been carried out with the aim of application of low-cost sensors for building monitoring, still there are some gaps in the literature that must be focused on in future research. The same systematic literature review ought to be carried by considering other databases (such as IEEE explore, science direct, and PubMed) to present comprehensive analysis of employed low-cost sensors for monitoring of building sector. It is also necessary to provide overview of the existing low-cost sensors in the market and literature for building monitoring including their application, their calibration techniques and comparison of information in catalogues, and their actual performance on the site against reference measurements.

**Author Contributions:** Conceptualization, B.M. and F.J.C.P.; methodology, F.L.-G.; investigation, R.P.S.; resources, F.L.-G.; data curation, B.M.; writing—original draft preparation, F.J.C.P.; writing— review and editing, B.M.; visualization, F.L.-G.; supervision, R.P.S.; project administration, R.P.S.; funding acquisition, B.M. All authors have read and agreed to the published version of the manuscript.

**Funding:** This research was funded by the Spanish Ministry of Economy and Competitiveness (grant number BIA2013-47290-R, BIA2017-86811-C2-1-R, and BIA2017-86811-C2-2-R), the Universidad de Castilla La Mancha (grant number 2018-COB-9092).

**Institutional Review Board Statement:** Not applicable.

**Informed Consent Statement:** Not applicable.

**Data Availability Statement:** No new data were created or analyzed in this study. Data sharing is not applicable to this article.

**Acknowledgments:** The authors are indebted to the Spanish Ministry of Economy and Competitiveness for the funding provided through the research projects BIA2017-86811-C2-1-R, and BIA2017-86811-C2-2-R founded with FEDER funds. It is also to be noted that funding for this research has been provided to Behnam Mobaraki by the Spanish Ministry of Economy and Competitiveness through its program for his Ph.D. It is also to be noted that part of this work was performed through grant number 2018-COB-9092 from Universidad de Castilla-La Mancha (UCLM).

**Conflicts of Interest:** The authors declare no conflict of interest.

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
