# Peer review of "Application of Low-Cost Sensors for Building Monitoring: A Systematic Literature Review"

_buildings, doi:10.3390/buildings11080336_

Round 1

Reviewer 1 Report

REVIEW

on article

Application of low-cost sensors for building monitoring: A systematic literature review

SUMMARY.

The article is a review of the literature on the use of low-cost sensors for building monitoring. The article contains a lot of useful information about the types of monitoring and the sensors used, which can greatly facilitate the reader's analysis and applicability for various tasks.

For internal monitoring, sensors are considered for measuring the efficiency of electrical appliances, analyzing emissions of pollutants and aerosol particles, thermal comfort, measuring humidity, and others.

Structural monitoring of buildings is presented by the analysis of vibration accelerations and deformations.

The final part of the article is presented by studying microcontrollers and communication protocols in the literature.

Comments:

  1. The Review considered only sensors used in the monitoring process. In my opinion, without analyzing the methodology for using certain sensors, the article loses significantly. The methodology of signal processing defines the need for the use of appropriate sensors. This is not covered in the review.
  2. Currently, BIM technologies are used in many countries, which are associated with monitoring. This opens up wide opportunities for monitoring important buildings. However, the article does not pay due attention to this.
  3. Line 70. "... The development of a building monitoring system (BMS) requires the installation of a data acquisition systems for real-time observation of parameters and a network for storing of the data". Data collection is only the first stage. Monitoring is a system of continuous observation of phenomena and processes occurring in the environment (or society), the results of which serve to justify management decisions to ensure the safety of people and objects. In terms of management decisions, the article does not pay attention.
  4. Line 97-100. There are many of these sensors on the market. To monitor massive buildings, sensors are needed that work correctly in the 0.01 - 10 Hz range. If possible, pay attention to low-frequency sensors.
  5. In terms of structural monitoring methodology, I recommend that you read the article and, if possible, refer to Lyapin, A.; Beskopylny, A .; Meskhi, B. Structural Monitoring of Underground Structures in Multi-Layer Media by Dynamic Methods. Sensors 2020, 20, 5241. https://doi.org/10.3390/s20185241

In general, the article is devoted to the relevant and interesting problem of using low-cost sensors in building monitoring tasks. Therefore, the article can be an interesting Review that will attract the attention of readers.

However, there are many comments, so I recommend the article for publication after major corrections.

Author Response

EDITOR/REFEREE(S)' COMMENTS TO AUTHOR:

The authors sincerely appreciate the positive comments of the Reviewer and are very grateful for their suggestions and observations. These comments certainly have improved the quality of the paper. In addition, they gave us useful hints on our future research. Detailed responses, to each of the reviewer’ comments are provided in the following lines. The answers of the authors are highlighted in green and the added new parts are highlighted in red. In the Revised version of the article, the changes have been highlighted by blue.

REVIEWER: 1

1- The Review considered only sensors used in the monitoring process. In my opinion, without analyzing the methodology for using certain sensors, the article loses significantly. The methodology of signal processing defines the need for the use of appropriate sensors. This is not covered in the review

We appreciate this comment, in the case of the work presented it was about analyzing low-cost sensors, to present the methodologies for data processing of low-cost monitoring systems, the authors have added a new subsection “2.2. Data processing of low-cost monitoring systems” to the article (from line 307 to 338). Low-cost monitoring systems refer to the metering systems that are based on low-cost sensors (LCSs) and microcontrollers, which are the main focus of this article.  The main characteristic of these systems is that they contain open-source programs which permits the user to have access to the library of the installed low-cost sensors, copying, improving, and developing an algorithm for post-processing of the sensors’ measurements.

Even though these groups of the metering devices are not the focus of the authors in this article, to help the reader distinguish the difference between data processing of the low-cost sensors and conventional ones, a brief explanation of them was provided. These devices are normally based on the close source platforms and have already programmed by the associated companies. This implies that only the original authors are able to inspect and improve the developed software. These companies provide the users by a software for post-processing of information and the users will receive periodically the update of the source code. In the corrected version of the paper, the following subsection has been added:

2.2. Data processing of low-cost monitoring systems

New achievements in technology of metering devices, wireless communications, and data processing methods, in conjunction with ever increasing number of aged buildings, have led to development of more efficient monitoring systems. In fact, the principal feature of the current low-cost monitoring devices is that they contain open-source programs. These metering systems provide permission to the users to use the library source of the utilized low-cost sensors, content of the data acquisition system, and criteria for the processing of the measurements. This implies that the users have chance of copying, improving and developing an algorithm for post-processing of their own measurement data, freely [81]. In the context of building energy retrofitting/monitoring various scholars developed specific data acquisition systems [87,94,95]. The challenges of simulations, experimental measurements, and post processing of data in terms of building thermal monitoring was studied by Evangelisti et al [96]. In the first step, the authors investigated how the disturbing factors might influence the final results of U-value and then they developed a novel post-processing method data analysis called “linear trendline” approach [96]. Taking into account the finite element simulation and in-situ measurements, they received satisfactory results by reducing the difference between the measured and calculated transmittance value. Marquez et al. developed a novel low-cost U-value meter which contains open source software and incorporates a user interface [97]. It also delivers real time monitoring of the data, processes of the information, detecting communication errors and warns of outliers. Example of building energy management system using self-deployed open source platforms for data processing of multiple variables can be found in [98]. Some authors, in this line, have introduced an open source signal processing algorithm in terms of monitoring structural parameters of buildings [52,59,87,99].

However, majority of the conventional devices in the market in any fields of building monitoring, have already programed by the associated companies and they are based on the close source platforms. This means that only the original authors are able to inspect and improve the developed software. These companies provide the users by the software for post-processing of information and the users will receive the update of the source code, annually. The users must agree that they cannot alter or copy anything from the source code. Examples of these metering devices for measuring thermal transmittance of the building envelope can be find in [80] and [18].

Since a new section has been introduced, we have changed a paragraph in the introduction (from line 237 to line 249), from:

This paper is organized as follows. In the section 2, the methods applied for inclusion, exclusion and classification of the articles are presented and specific objectives of the research are defined. In this section also some research questions are presented to make the systematic review and follow the determined protocol. In section 3, detailed explanation about various fields of building monitoring and reviewed article are given. In the section 4, presentation of the different types of microcontrollers and communication protocols in the literature are review and the preferred ones for installation of low-cost monitoring devices are presented. In section 5, the obtained results, importance of different aspects of building monitoring are discussed and some proposal for future investigation are presented, Finally, in section 6 some conclusions are drawn.

 to:

This paper is organized as follows. In the section 2, the methods applied for inclusion, exclusion and classification of the articles are presented and specific objectives of the research are defined. Moreover, the methodology for data processing of the low-cost monitoring devices is introduced. In this section also some research questions are presented to make the systematic review and follow the determined protocol. In section 3, detailed explanation about various fields of building monitoring and reviewed article are given. In the section 4, presentation of the different types of microcontrollers and communication protocols in the literature are review and the preferred ones for installation of low-cost monitoring devices are presented. In section 5 significance of combination of BIM and LCSs are introduced, and the associated references are given. Section 6, the obtained results, importance of different aspects of building monitoring are discussed and some proposal for future investigation are presented, Finally, in section 7 some conclusions are drawn.

2- Currently, BIM technologies are used in many countries, which are associated with monitoring. This opens up wide opportunities for monitoring important buildings. However, the article does not pay due attention to this.

The authors agree with the comment of the Reviewer that the highlighted idea was not completely explained. To fill this gap a new section “5. Integration of BIM and low-cost sensors” has been added to the modified version of the article (from line 776-824), introducing integration of BIM and LCSs. This new section presents growing interest on integration of BIM and LCSs in the SCOPUS database, importance of this idea, and some examples from the mentioned database. In the corrected version of the paper, the following section has been added:

  1. Integration of BIM and low-cost sensors.

This section investigates examples of articles dealing with integration of LCSs and BIM. The first concept of BIM was proposed in a book entitled “Building Product Models” written by Eastman in 1993 [204]. This book introduces principal idea of architectural information modeling, linked model components, and information changes. The growing interests in integration of LCSs and BIM in SCOPUS database is illustrated in figure 12. It shows that connection of these two technologies is significantly growing with time. For example, this is indicated by the number of total publications on this topic, which it has grown from 1 in 2011 to 58 in 2020.

Figure 12. Number of works addressing integration of BIM and LCS in the Scopus database from 2011 to 2020.

There are diverse research lines in the literature dealing with application of BIM to low-cost monitoring of buildings. Some of the authors focused on the connection of low-cost data acquisiton system to BIM model [205]. In fact, wise selection of the methodology for transfering the data from sensors to BIM model may help the users to make real-time monitoring of the measurements, accurate inclusion of geometric shapes and associated data, and finaly avoiding the loss-data. Kensek inevstigated feasibility of connecting environmental sensors such as light, humidity and  to BIM. To do so, they have tried to make the Arduino-Revit connection through Dynamo, Rhino and Grasshopper [206]. Chang et al. described a methodology for visualizing measurements recorded with sensors in BIM models, employing several perspectives in which to support complex decisions that require interdisciplinary information. They utilized Dynamo to bring procedural information from Arduino into 3D BIM. Moreover, they focused on design of a platform for the connection of low-cost sensors to various microcontrollers [Arduino and Raspberyy Pi] [207]. Gunduz et al. presented a sample software architecture for integration of BIM, geographical information system [GIS] and IoT for supporting comfort analysis [208]. The authors used Arduino UNO to collect measurements from LCSs of temperature, humidity and light level. Shahinmoghadam et al. studied the synergistic benefits of BIM, IoT and virtual reality [VR] for estblishing a low-cost monitoring thermal comfort system [209].

However, the other group of authors have been mainly focused on integration of BIM and LCSs to support visualization of live data streams for monitoring various parameters of buildings [210]. An Arduino-based data acquisition system developed by Natephra and Motamedi for real-time monitoring of thermal comfort parameters such as: indoor temperature, humidity and light intensity [211]. Desogus et al. developed a BIM platform for real-time monitoring of the data [temperature and luminance] coming from low-cost sensors within parametric model of a historical building [212]. Li et al. proposed an automatic as-built BIMs framework that transforms the noisy 3D point cloud generated by low-cost RGB-D sensor [213]. Shen and Lu proposed a new methodology to engage the BIM model as a control system of building energy performance. They designed a parametric adaptive skin system [PASS] to combine the adaption of natural sunlight use for higher building performance [214]. Rahmani Asl et al. developed a framework for BIM-based performance optimization. They used BPOpt in reducing the energy consumption while increasing the sufficient daylight level for buildings [215]. Teizer et al. focused on permanent availability of actual performance data sets throught IoT that combines environmental information in a cloud-based BIM platform [216].

Moreover, two articles related to the literature review of BIM have been added to the section 1 (Introduction, from line 188 to 191). In the corrected version of the paper, the following text is as follow:

There are various review papers in the literature concerning: the approaches for smart monitoring of buildings [76], environmental monitoring sensors in buildings [77], heritage building information modeling [78], BIM-based end-of-lifecycle decision making and digital deconstruction [79]

The added references are as follows:

[78]. López FJ, Lerones PM, Llamas J, Gómez-García-Bermejo J, Zalama E. A review of heritage building information modeling (H-BIM). Multimodal Technol Interact. 2018;2(2).

[79]. Akbarieh A, Jayasinghe LB, Waldmann D, Teferle FN. BIM-based end-of-lifecycle decision making and digital deconstruction: Literature review. Sustain. 2020;12(7).

Since a new section has been introduced, we have changed a paragraph in the introduction (from line 237 to line 249), from:

This paper is organized as follows. In the section 2, the methods applied for inclusion, exclusion and classification of the articles are presented and specific objectives of the research are defined. In this section also some research questions are presented to make the systematic review and follow the determined protocol. In section 3, detailed explanation about various fields of building monitoring and reviewed article are given. In the section 4, presentation of the different types of microcontrollers and communication protocols in the literature are review and the preferred ones for installation of low-cost monitoring devices are presented. In section 5, the obtained results, importance of different aspects of building monitoring are discussed and some proposal for future investigation are presented, Finally, in section 6 some conclusions are drawn.

 to:

This paper is organized as follows. In the section 2, the methods applied for inclusion, exclusion and classification of the articles are presented and specific objectives of the research are defined. Moreover, the methodology for data processing of the low-cost monitoring devices is introduced. In this section also some research questions are presented to make the systematic review and follow the determined protocol. In section 3, detailed explanation about various fields of building monitoring and reviewed article are given. In the section 4, presentation of the different types of microcontrollers and communication protocols in the literature are review and the preferred ones for installation of low-cost monitoring devices are presented. In section 5 significance of combination of BIM and LCSs are introduced, and the associated references are given. Section 6, the obtained results, importance of different aspects of building monitoring are discussed and some proposal for future investigation are presented, Finally, in section 7 some conclusions are drawn.

3- Line 70. "... The development of a building monitoring system (BMS) requires the installation of a data acquisition systems for real-time observation of parameters and a network for storing of the data". Data collection is only the first stage. Monitoring is a system of continuous observation of phenomena and processes occurring in the environment (or society), the results of which serve to justify management decisions to ensure the safety of people and objects. In terms of management decisions, the article does not pay attention

The authors agree with the comment of the Reviewer that the highlighted sentence was not sufficiently explained. To solve this issue, in the corrected version of the article the highlighted text has been changed from:

The development of a building monitoring system (BMS) requires the installation of a data acquisition system for real time observation of parameters and a network for storing of the data [19]. Unfortunately, due to the complexity of the sensors, traditional BMS is not an easy task as it entails specialized programing and maintenance of the system. In this way various monitoring systems have been produced by companies for direct supervision of building performance. As depicted in figure 1, a common monitoring system contains three main parts. (1) A detection part. This part contains sensor or data acquisition system to measure the events and changes of a parameter. (2) data transfer system. This part is used to transmit readings from one place to another through a communication method and (3) storage part. This part consists of a component to keep the digital data.

to:

The development of a building monitoring system (BMS) requires the installation of a data acquisition system for real time observation of parameters and a network for storing of the data [20]. Unfortunately, due to the complexity of the sensors, traditional BMS is not an easy task as it entails specialized programing and maintenance of the system. In this way various monitoring systems have been produced by companies for direct supervision of building performance. As depicted in figure 1, a common monitoring system contains three main parts. [1] A detection part. This part contains sensor or data acquisition system to measure the events and changes of a parameter. [2] data transfer system. This part is used to transmit readings from one place to another through a communication method and [3] storage part. This part consists of a component to keep the digital data.

After saving and processing the data, the principal objective is to assess the workability of the building and safety of the habitants. Recently, diverse technologies have been developed with the purpose of real time monitoring of buildings and safety management of people. In fact, the most flourishing technology for this aim is BIM. This tool is a novel approach to monitor, analyze, design, and safety management, wherein visualization of the building state is utilized to enable the exchange and interoperability of data. BIM has been adopted by CAD software such as: Autodesk Revit [21], ArchiCAD [22] and Allplan [23]. In 1900s Matilla et al. studied significance of studying the connections between management and safety [24]. Research of Ding et al. indicated that the quantity of the published articles on BIM from the perspective of the safety management is 7%. The other advanced technology that brought great potential for safety management of people is wireless sensor network (WSN). With cheap, low-power consumption, and dynamic networking characteristics, WSNs are good option for collection of environmental and structural information of buildings and return it to visualizing unit for safety management and risk analysis [25]. Cheung at al. integrated BIM and WSN into a single system to visualize the construction site, monitor the safety level through an interface and remove the risk/danger of the gas automatically [26]. They have also carried out safety management process of underground structures by monitoring the other elements of environmental parameters such as temperature and humidity. They installed the sensor nodes at different location of the structure, and in any area where an abnormal situation is realized, the BIM model alarms the area and ventilator starts working, automatically.

3- Line 97-100. There are many of these sensors on the market. To monitor massive buildings, sensors are needed that work correctly in the 0.01 - 10 Hz range. If possible, pay attention to low-frequency sensors.

The authors completely agree with the Reviewer about the interest of applying the proposed information to the article. Therefore, to fill this gap we have added a paragraph presenting a set of low frequency accelerometers in the literature. Therefore, in the corrected version of the paper, the following text has been changed from (from the line 145 to 173):

In terms of structural monitoring, there are various tools to measure different parameters such as acceleration, strain and stresses. Accelerometer is an electromechanical device that determines the acceleration forces imposed to or acting on an object. There are varieties of accelerometer that work based on different mechanism such as: (1) piezoelectric, this type of accelerometer uses piezoelectric effect of specific material and transform a type of energy into another one and produce electrical signal in response to parameter is being measured. (2) piezoresistive, this type of accelerometer generates resistance changes in displacement sensor which is part of accelerometer system. This type of accelerometer is the best option for measuring impulse where the amplitude and frequency range are high. (3) capacitive micro electro mechanical system (MEMS), for construction of this type of accelerometer, MEMS technology is being used and works according to capacitance changes in a seismic mas under acceleration. Table 2 provides examples of the mentioned types of accelerometers in the market providing the mechanism, the name, acceleration range, frequency range, and the references in the literature.

to:

In terms of structural monitoring, there are various tools to measure different parameters such as acceleration, strain and stresses. Accelerometer is an electromechanical device that determines the acceleration forces imposed to or acting on an object. There are varieties of accelerometer that work based on different mechanism such as: [1] piezoelectric, this type of accelerometer uses piezoelectric effect of specific material and transform a type of energy into another one and produce electrical signal in response to parameter is being measured. [2] piezoresistive, this type of accelerometer generates resistance changes in displacement sensor which is part of accelerometer system. This type of accelerometer is the best option for measuring impulse where the amplitude and frequency range are high. [3] capacitive micro electro mechanical system (MEMS), for construction of this type of accelerometer, MEMS technology is being used and works according to capacitance changes in a seismic mas under acceleration. Table 2 provides examples of the mentioned types of accelerometers in the market (including both and high frequency) providing the mechanism, the name, acceleration range, frequency range, and the references in the literature. The accelerometers utilized for measuring specific human activities, transportation and mechanical devices must be developed specially for low frequency and high sensitivity, range from 1 to 10 Hz [40]. Tian et al. developed, a piezoelectric accelerometer on n-type single crystal silicon and examined the sensor in terms of maximum stress, natural frequency, and output voltage under an acceleration through finite element method. The sensitivity of the developed accelerometer was 9 , the linearity was 0.0205, and the hysteresis was 0.0033 [41]. Another example of low frequency piezoelectric accelerometers in the literature are [42,43]. Liu et al. proposed a novel low frequency fiber bragg grating (FBG) accelerometer with a bended spring plate. The experiments showed the sensitivity of the accelerometer was more than 1000  when the frequency is within the 0.7 to 20 Hz [44]. Another example of low frequency FBG accelerometers can be found in [45–47]. Zhu et al. designed high resolution, low frequency and low-noise tri-axial digital MEMS accelerometer for monitoring large-scale civil infrastructures [48]. Examples of the other low frequency MEMS accelerometers were presented in [49,50].

4- In terms of structural monitoring methodology, I recommend that you read the article and, if possible, refer to Lyapin, A.; Beskopylny, A .; Meskhi, B. Structural Monitoring of Underground Structures in Multi-Layer Media by Dynamic Methods. Sensors 2020, 20, 5241. https://doi.org/10.3390/s20185241

The authors have read the highlighted paper and added to the article. Therefore, line 27 to 43 of the article has been changed from:

Accordingly, engineers decided to control performance of the buildings by monitoring two main parameters: 1- Structural parameters: in terms of structural system identification (SSI), some numerical approaches have been developed for inferring mechanical parameters of structures modeled with 1D elements (such as steel and concrete buildings, cable stayed bridges, trusses and frames) [3–5], 2D elements (such as tunnels, culverts, and dams) [6], and 3D elements [7]. A review of research carried out in the field of SSI was presented by Sirca Jr. and Adeli [8].

to:

Accordingly, engineers decided to control performance of the buildings by monitoring two main parameters: 1- Structural parameters: in terms of structural system identification (SSI), some numerical approaches have been developed for inferring mechanical parameters of structures modeled with 1D elements (such as steel and concrete buildings, cable stayed bridges, trusses and frames) [3-5], 2D elements [such as tunnels, culverts, and dams] [6], and 3D elements [7]. In terms of deriving parameters of structures modelled with 3D elements, various approaches have been presented in the literature [8,9]. For instance, Mobaraki and Vaghefi developed 3D finite element models to measure peak pressure and the peak particle velocity (PPV), at critical points of 4 cross-sectional shape of tunnel [box shape, circular shape, horseshoe shape, and semi-ellipse shape] [10]. Lyapin et al. introduced a methodology for monitoring buried structures of arbitrary cross-section, located in layered media, affected by different external dynamic loads [11]. They developed and analytical approach for determination of resonance zones and dynamic response of structures. In addition, they conducted an experimental study for measuring acceleration at the ceiling center point of an underground pedestrian road in Russia. A review of research carried out in the field of SSI was presented by Sirca Jr. and Adeli [12].

The added reference is as follow:

[10]- Lyapin A, Beskopylny A, Meskhi B. Structural monitoring of underground structures in multi-layer media by dynamic methods. Sensors (Switzerland). 2020;20(18):1–19.

Reviewer 2 Report

The paper focuses on the literature review of low cost sensors in building applications. The methodology is clear. I have just a doubt on the literature review on the database Scopus, as not all the paper published in mdpi appears in the bibliography. Journal as Sensors, Electronics and Energies are really important on this topic. For a smoke an Europea Project name Hello! Is based on the use of low cost sensors for the hydrothermal monitoring of the building envelope. Two paper have been published on it: https://doi.org/10.3390/electronics8060643 and https://doi.org/10.3390/en13112950. It is strange that you didn’t find these papers. Also the Journal Sensors has a special issue on this topic https://www.mdpi.com/journal/sensors/special_issues/SFCHM. Probably the key words must enlarged or the database is not complete. Try to add these research and consideration. The conclusions are completely not adeguate for this paper. Improve the comments with more details useful for the scientific community.

Author Response

EDITOR/REFEREE(S)' COMMENTS TO AUTHOR:

The authors sincerely appreciate the positive comments of the Reviewer and are very grateful for their suggestions and observations. These comments certainly have improved the quality of the paper. In addition, they gave us useful hints on our future research. Detailed responses, to each of the reviewer’ comments are provided in the following lines. The answers of the authors are highlighted in green and the added new parts are highlighted in red. In the revised version of the article, the changes have been highlighted by purple.

REVIEWER: 2

1- The paper focuses on the literature review of low cost sensors in building applications. The methodology is clear. I have just a doubt on the literature review on the database Scopus, as not all the paper published in mdpi appears in the bibliography. Journal as Sensors, Electronics and Energies are really important on this topic.

The authors completely agree with the comment of the Reviewer, as all the publications in mdpi do not appear in SCOPUS database. However, we searched in the aforementioned journals, we found very useful publications, and we added them to the article. Thus, 16 articles from the mdpi journals (Sensors, sustainability, Energies, Electronics, and Applied science) have been added to the modified version of the article. The added references are as follows:

[11]. Lyapin A, Beskopylny A, Meskhi B. Structural monitoring of underground structures in multi-layer media by dynamic methods. Sensors (Switzerland). 2020;20(18):1–19.

[13]. Bianconi F, Salachoris GP, Clementi F, Lenci S. A genetic algorithm procedure for the automatic updating of fem based on ambient vibration tests. Sensors (Switzerland). 2020;20(11):1–17.

[14]. Markiewicz J, Łapiński S, Kot P, Tobiasz A, Muradov M, Nikel J, et al. The quality assessment of different geolocalisation methods for a sensor system to monitor structural health of monumental objects. Vol. 20, Sensors (Switzerland). 2020. 1–39 p.

[15]. Anaf W, Cabal A, Robbe M, Schalm O. Real-time wood behaviour: The use of strain gauges for preventive conservation applications. Sensors (Switzerland). 2020;20(1).

[26]. Cheung WF, Lin TH, Lin YC. A real-time construction safety monitoring system for hazardous gas integrating wireless sensor network and building information modeling technologies. Sensors (Switzerland). 2018;18(2).

[41]. Tian B, Liu H, Yang N, Zhao Y, Jiang Z. Design of a piezoelectric accelerometer with high sensitivity and low transverse effect. Sensors (Switzerland). 2016;16(10).

[44]. Liu F, Dai Y, Karanja JM, Yang M. A low frequency FBG accelerometer with symmetrical bended spring plates. Sensors (Switzerland). 2017;17(1).

[48]. Zhu L, Fu Y, Chow R, Spencer BF, Park JW, Mechitov K. Development of a high-sensitivitywireless accelerometer for structural health monitoring. Sensors (Switzerland). 2018;18(1):1–16.

[50]. Sabato A, Feng MQ. Feasibility of frequency-modulated wireless transmission for a multi-purpose MEMS-based accelerometer. Sensors (Switzerland). 2014;14(9):16563–85.

[75]. Villacorta JJ, Del-val L, Martinez R., Balmori J., Magdaleno A, Lopez G, et al. Design and Validation of a Scalable, Reconfigurable and Low-Cost Structural Health Monitoring System. Sensors (Basel). 2021;21:648.

[97]. Andújar Márquez JM, Martínez Bohórquez MÁ, Gómez Melgar S. A New Metre for Cheap, Quick, Reliable and Simple Thermal Transmittance (U-Value) Measurements in Buildings. Sensors (Basel). 2017 Sep 3;17(9).

[157]. Andreotti M, Calzolari M, Davoli P, Pereira LD, Lucchi E, Malaguti R. Design and construction of a new metering hot box for the in situ hygrothermal measurement in dynamic conditions of historic masonries. Energies. 2020;13(11).

[158]. Lucchi E, Pereira LD, Andreotti M, Malaguti R, Cennamo D, Calzolari M, et al. Development of a compatible, low cost and high accurate conservation remote sensing technology for the hygrothermal assessment of historic walls. Electron. 2019;8(6).

[207]. Chang KM, Dzeng RJ, Wu YJ. An automated IoT visualization BIM platform for decision support in facilities management. Appl Sci. 2018;8(7).

[212]. Desogus G, Quaquero E, Rubiu G, Gatto G, Perra C. Bim and iot sensors integration: A framework for consumption and indoor conditions data monitoring of existing buildings. Sustain. 2021;13(8).

[213]. Li Y, Li W, Tang S, Darwish W, Hu Y, Chen W. Automatic indoor as-built building information models generation by using low-cost RGB-D sensors. Sensors (Switzerland). 2020;20(1).

2- For a smoke an Europe Project name Hello! Is based on the use of low cost sensors for the hydrothermal monitoring of the building envelope. Two papers have been published on it: https://doi.org/10.3390/electronics8060643 and https://doi.org/10.3390/en13112950. It is strange that you didn’t find these papers.

The authors agree that the proposed articles are so useful and are connected with the idea of this article. Therefore, the 2 mentioned references have been added to the modified version of the article (from line 562 to 566) as follows:

Andreottei et al. developed a novel hot box to derive thermal parameters of masonry and historic buildings, such as heat flux, surface temperature, relative humidity, and air temperature [157]. The main feature of the developed metering device are: reliability of the monitoring system, conservation of cultural heritage, ease of equipment installation, and finally economic cost [158].

The added references are as follows:

[157]. Andreotti M, Calzolari M, Davoli P, Pereira LD, Lucchi E, Malaguti R. Design and construction of a new metering hot box for the in situ hygrothermal measurement in dynamic conditions of historic masonries. Energies. 2020;13(11).

[158]. Lucchi E, Pereira LD, Andreotti M, Malaguti R, Cennamo D, Calzolari M, et al. Development of a compatible, low cost and high accurate conservation remote sensing technology for the hygrothermal assessment of historic walls. Electron. 2019;8(6).

3- Also the Journal Sensors has a special issue on this topic https://www.mdpi.com/journal/sensors/special_issues/SFCHM. Probably the key words must enlarged or the database is not complete. Try to add these research and consideration.

The authors agree with comment of reviewer and the articles of this special issue were missed in the paper. The authors have checked this special issue “Sensors for cultural heritage monitoring” and added 4 related references of this special issue to the article. Therefore, the below text has been added to the correct version of the article (from line 43 to 50):

The methodology for identification of modal parameters in historical building using stochastic subspace identification algorithm carried out by Bianconi et al. [13]. Markiewicz et al. utilized geolocalisation approach to specify the location of the structural monitoring system that allow to georeferenced the measurements carried out by the sensors. This analysis is useful for data processing related to the monitored structure and its features [14]. Applicability of strain gauge to real time monitoring of wood behavior carried out was carried out by Anaf et al. in a church where a new heating system was installed [15].

The fourth article ([75]) has been added to the line 187 of the article. The added references are as follows:

[13]. Bianconi F, Salachoris GP, Clementi F, Lenci S. A genetic algorithm procedure for the automatic updating of fem based on ambient vibration tests. Sensors (Switzerland). 2020;20(11):1–17.

[14]. Markiewicz J, Łapiński S, Kot P, Tobiasz A, Muradov M, Nikel J, et al. The quality assessment of different geolocalisation methods for a sensor system to monitor structural health of monumental objects. Vol. 20, Sensors (Switzerland). 2020. 1–39.

[15]. Anaf W, Cabal A, Robbe M, Schalm O. Real-time wood behaviour: The use of strain gauges for preventive conservation applications. Sensors (Switzerland). 2020;20(1).

[75]. Villacorta JJ, Del-val L, Martinez R., Balmori J., Magdaleno A, Lopez G, et al. Design and Validation of a Scalable, Reconfigurable and Low-Cost Structural Health Monitoring System. Sensors (Basel). 2021;21:648.

4- The conclusions are completely not adequate for this paper. Improve the comments with more details useful for the scientific community.

The authors agree with the Reviewer that the conclusions were not properly presented. To solve this issue the conclusions have been modified using bullet type statements for a better understanding changing from:

In this article, a systematic literature review of application of low-cost sensors for building monitoring has been presented. After implementation of inclusion and exclusion process of the founded articles in SCOPUS database, 99 linked articles from 2006 to 2020 were chosen and discussed. According to the reviewed articles, we could provide the readers a general overview of the parameters to be monitored in building sector, including indoor field (electricity consumption, air quality and thermal comfort & HVAC) and structural field (vibration and strain). It was observed that majority of the authors dedicated their studies for indoor monitoring of buildings (62.6%). However, among the reviewed articles for indoor monitoring, we could find only a single group of authors presenting low-cost approach for assessing thermal/energy performance of buildings. This field of research should be addressed more in future as determining thermal parameters of buildings (e.g., transmittance and resistance values) can specify the level of energy scape and also helps engineers for energy retrofitting of buildings’ envelopes. When it comes evaluating low-cost monitoring of indoor air quality, it has been found that all the authors considered only on aerosol and  detection in historical, residential and educational building. Future studies should focus on developing low-cost monitoring systems for detection of , , , , fungi and bacteria in schools. In the case of hospital researchers should focus on measuring the level of  ,  and . In the case of administrative buildings future studies must be carried out for monitoring of , ,  and . In the case of residential buildings, it is also essential to focus on monitoring the  and  level.

Regarding the architecture of the low-cost data acquisitions systems, two tables providing list of utilized microcontrollers (17 types) and communication protocols (8 types) in the literature were drawn. It has been discovered that Arduino microcontroller and ZigBee communication protocol have attracted a great deal of attention. It is important to highlight that when deploying low-cost devices for building monitoring, it is essential to determine the influential factors and also enhance the precision of monitoring while holding the simplicity of the operational principals.

This article indicates that although remarkable number of studies has been carried out with the aim of application of low-cost sensors for building monitoring, still there are some gaps in the literature that must be focused on in future research. The same systematic literature review ought to be carried by considering other databases (such as IEEE explore, science direct, and PubMed) to present comprehensive analysis of employed low-cost sensors for monitoring of building sector. It is also necessary to provide overview of the existing low-cost sensors in the market/literature for building monitoring including their application, their calibration techniques and comparison of information in catalogues and their actual performance on the site against reference measurements.

The search for papers was limited to the combination of keywords. Further limitations lay in bias risk assessment factors, which were not considered in the included articles in the literature review performed in this research.

to:

In this article, a systematic literature review of application of low-cost sensors for building monitoring has been presented. After implementation of inclusion and exclusion process of the founded articles in SCOPUS database, 99 linked articles from 2006 to 2020 were chosen and discussed. According to the reviewed articles, we could provide the readers an overview of the parameters to be monitored in building sector, including indoor field (electricity consumption, air quality and thermal comfort & HVAC) and structural field (vibration and strain). In the case of assessing the thermal comfort and satisfaction of buildings’ habitants, it was found out that majority of the authors (58.3%) focused their studies on monitoring of the said parameters in public buildings. This is due to the fact that, the people spend 90% of the life inside buildings. In addition, the economic motivations for spending money in building sector to raise productivity of working staff is unquestionable. As it has proven that attenuation of working performance by 0.5-5% causes a loss of 20 to 200 billion dollars per year. Reviewing of the articles for low-cost monitoring of the indoor air quality indicated that 41.7% of scholars were focusing on implementation of their devices on monitoring of various types of buildings such as educational, historical, residential and laboratory. Whereas the annual saving of 6 to 19 billion dollars comes from reduction of respiratory disease. Therefore, future studies should focus on developing low-cost monitoring systems for detection of , , , , fungi and bacteria in schools. In the case of hospital researchers should focus on measuring the level of  ,  and . In the case of administrative buildings future studies must be carried out for monitoring of , ,  and . In the case of residential buildings, it is also essential to focus on monitoring the  and  level.

Regarding the architecture of the low-cost data acquisitions systems, two tables providing list of utilized microcontrollers (17 types) and communication protocols (8 types) in the literature were drawn. It has been discovered that Arduino microcontroller and ZigBee communication protocol have attracted a great deal of attention. It is important to highlight that when deploying low-cost devices for building monitoring, it is essential to determine the influential factors, and also enhance the precision of monitoring while holding the simplicity of the operational principals. When it comes appraising the methodologies for data processing of the low-cost monitoring systems in the literature, it has been found that majority of the authors have used open-source models that are peer production and encourage open collaboration. This approach led to massively reduction of the monitoring costs in the literature, as the source codes/libraries/software are freely available for feasible modifications and redevelopment by any users. In addition, integration of low-cost monitoring devices and BIM model has been studied and growing interests on this field has been analyzed.

This article reviewed several attempts for designing of platforms to receive raw sensors data as an input and delivers automatic visualization of the processed data in BIM. In fact, the most common platform in the literature for connection of microcontrollers and BIM model is Dynamo. Reviewing of the other articles indicated real-time streaming of the sensor data is not limited only to BIM technology, as IoT and VR has been implemented in various low-cost monitoring projects to increase the project productivity.

This article indicates that although remarkable number of studies has been carried out with the aim of application of low-cost sensors for building monitoring, still there are some gaps in the literature that must be focused on in future research. The same systematic literature review ought to be carried by considering other databases (such as IEEE explore, science direct, and PubMed) to present comprehensive analysis of employed low-cost sensors for monitoring of building sector. It is also necessary to provide overview of the existing low-cost sensors in the market/literature for building monitoring including their application, their calibration techniques and comparison of information in catalogues and their actual performance on the site against reference measurements.

Round 2

Reviewer 1 Report

All my comments were taken into account by the authors. The article has been significantly revised and now is a large-scale review devoted to an urgent problem. I recommend the article for publication

Reviewer 2 Report

The literature review is impressive